# Integration of IL-2 and IL-4 signals coordinates divergent regulatory T cell responses and drives therapeutic efficacy

**Julie Y Zhou, Carlos A Alvarez, Brian A Cobb\***

Department of Pathology, Case Western Reserve University School of Medicine, Cleveland, United States

**Abstract** Cells exist within complex milieus of communicating factors, such as cytokines, that combine to generate context-specific responses, yet nearly all knowledge about the function of each cytokine and the signaling propagated downstream of their recognition is based on the response to individual cytokines. Here, we found that regulatory T cells (Tregs) integrate concurrent signaling initiated by IL-2 and IL-4 to generate a response divergent from the sum of the two pathways in isolation. IL-4 stimulation of STAT6 phosphorylation was blocked by IL-2, while IL-2 and IL-4 synergized to enhance STAT5 phosphorylation, IL-10 production, and the selective proliferation of IL-10-producing Tregs, leading to increased inhibition of conventional T cell activation and the reversal of asthma and multiple sclerosis in mice. These data define a mechanism of combinatorial cytokine signaling and lay the foundation upon which to better understand the origins of cytokine pleiotropy while informing improved the clinical use of cytokines.

**\*For correspondence:**
brian.cobb@case.edu

**Competing interests:** The authors declare that no competing interests exist.

## Introduction

IL-2 and IL-4 are both pleiotropic cytokines and members of a family in which γC (CD132) is a shared receptor subunit. These and many other cytokines have undergone intensive study because of their ubiquitous and essential roles in health and disease (*Leonard et al., 2019*), yet their functions can be highly divergent and context dependent.

IL-2 signals optimize effector and memory T cell responses as a part of normal immune protection (*Boyman and Sprent, 2012*; *Boyman et al., 2010*). From a pro-inflammatory perspective, IL-2 can drive the proliferation of activated CD4[+] and CD8[+] effector T cells (*Morgan et al., 1976*; *Liao et al., 2013*), enhance natural killer cell cytotoxicity (*Henney et al., 1981*; *Siegel et al., 1987*), and augment B cell proliferation and antibody secretion (*Mingari et al., 1984*). However, IL-2 is also potently anti-inflammatory, driving Fas-mediated activation-induced cell death (*Refaeli et al., 1998*) while promoting Treg survival (*Chinen et al., 2016*), with its loss resulting in severe autoimmunity and inflammation (*Horak, 1995*). Its pleiotropy can be partly explained by the strategic and context specific expression of its receptors, with IL-2 receptor α (IL-2Rα; CD25) being a common marker for activated conventional T cells (Tconv) as well as an established constitutive marker for Tregs (*Malek, 2008*; *Spolski et al., 2017*). CD25 is part of the high affinity receptor complex with IL-2 receptor β (IL-2Rβ; CD122) and γC, although it can function as a low-affinity receptor in isolation (*Waldmann, 1989*). An intermediate affinity IL-2R excluding CD25 is also known, and it consists of a heterodimer of IL-2Rβ and IL-2Rγ (*Boyman and Sprent, 2012*). The activity of IL-2 is dependent upon JAK/STAT signaling, with STAT3 and STAT5 being the primary mediators of its transcriptional regulation of target genes (*Johnston et al., 1995*).

IL-4 has perhaps an even broader impact and has two receptor forms. The type I IL-4 receptor is comprised of IL-4 receptor α (IL-4Rα; CD124) and γC, whereas the type II receptor contains IL-4Rα and IL-13 receptor α1 (IL-13Rα; CD213A1) (*Nelms et al., 1999*). Like many cytokines, signaling is

**eLife digest** The immune system is essential to defend our bodies from attacks of microbial invaders. It contains many types of cells that communicate with each other through proteins called cytokines. Cytokines act as messages that can promote or suppress the immune response. Since several types of immune cells can exist within the same environment, many different signals are sent simultaneously, from which each individual cell must tease out the correct message. So far, it has been unclear how cells decode simultaneous messages, which may be key for improving therapeutics for inflammatory diseases.

To investigate this, Zhou et al. studied the effect of two cytokines (IL-2 and IL-4) on a major type of suppressive immune cell called regulatory T cell or Treg. This revealed that when Tregs grown in the laboratory were simultaneously exposed to IL-2 and IL-4, the cytokine duo boosted the production of new Tregs much more than using either cytokine alone. Together, the cytokines also increased the production of another cytokine (IL-10), and over time, Treg cells producing IL-10 divided more frequently, leading to an even more robust ability to suppress overactive immune responses.

Zhou et al. then validated the experiments in cells using mouse models of asthma and multiple sclerosis. Treatment with an IL-2 and IL-4 cytokine cocktail not only prevented but reversed the symptoms of the diseases to a greater extent than either cytokine alone, suggesting that this could be a significantly improved therapy for patients affected by these conditions.

This study provides important insights into how cytokines can work together to substantially improve the activity of immune cells and reduce inflammation. It also introduces a promising new treatment strategy to dampen or even eliminate a range of devastating symptoms caused by inflammatory diseases.

mediated by the JAK/STAT pathway, with STAT6 being a main transcriptional mediator (*Hou et al., 1994*; *Gadani et al., 2012*). IL-4 is historically associated with allergy through its ability to stimulate class switching to IgE and expression of the IgE receptor (*Geha et al., 2003*), although it is also critical for parasite clearance (*Urban et al., 1991*) and differentiation of naïve T cells into a type two helper (Th2) phenotype (*Paul, 2015*), all of which are considered pro-inflammatory. Yet, IL-4 is also a key differentiating factor for wound healing macrophages (*Gordon, 2003*), which is essential for tissue repair, and is known to oppose IL-17-mediated inflammatory diseases like psoriasis and experimental autoimmune encephalomyelitis (EAE) (*Vogelgesang et al., 2010*; *Weigert et al., 2008*). Furthermore, IL-4R signaling has been shown to inhibit neutrophil effector functions in vivo, which pertains to inflammatory diseases where significant neutrophil infiltrates are found (*Woytschak et al., 2016*; *Impellizzieri et al., 2019*). In fact, IL-4 has been shown to elicit IL-10 in both macrophages and T cells (*Krzyszczyk et al., 2018*; *Mitchell et al., 2017*), helping to explain how IL-4 is also potently anti-inflammatory.

Although much is known about the signaling initiated by these and most other cytokines, understanding of how multiple cytokine signals integrate in complex environments remains limited. For many cytokines, the JAK/STAT cascades underlie their transcriptional impact (*Rawlings et al., 2004*), although a small number of JAK and STAT proteins belies the divergent nature of the responses. STAT homo- and hetero-multimerization is one mechanism that exponentially expands the available transcription outcomes (*O'Shea et al., 2015*). There is also evidence that cytokines can lead to the transcriptional enhancement or diminution of the signaling machinery for other cytokines (*Lin and Leonard, 2019*). For example, IL-2 stimulation drives the expression of IL-4Rα, thereby enhancing subsequent responses to IL-4 through serial activation of STAT5 and STAT6, respectively, ultimately leading to Th2 T cell differentiation (*Zhu et al., 2001*). This is additive signaling, whereby the canonical pathways act in concert to drive T cell differentiation, and follows our knowledge of each pathway in isolation. These mechanisms partially explain how a small number of STAT molecules can drive diverse cellular outputs, but do not account for the possibility of simultaneous stimulation that may occur in the complex array of cytokines that exist in any biological environment.

Here, we have discovered that the combination of IL-2 and IL-4 stimulation in Tregs culminates in a synergistic enhancement of Treg function and proliferation in vitro and in vivo. This effect is driven

by a uniquely integrated response requiring simultaneous rather than serial stimulation. These findings reveal a novel mechanism underlying cytokine signaling which can drive unexpected outcomes that are differential to the simple combination of pathways, thereby laying the groundwork for a more complete picture of cytokine pleiotropy and potentially leading to enhancements to the use of cytokines in the clinical setting.

## Results

### IL-2 and IL-4 synergistically promote IL-10 production by Tregs

Our previous findings demonstrated that IL-4 impacts IL-10 production in FoxP3$^+$ Tregs (*Jones et al., 2019*), but the nature of the functional interaction between IL-4 and IL-2, a key survival factor for Tregs in vitro and in vivo, remained obscure. Thus, we created a novel dual reporter mouse by crossing *Foxp3$^{RFP}$* (*Wan and Flavell, 2005*) and *Il10$^{GFP}$* (*Kamanaka et al., 2006*) mice, thereby enabling live sorting of FoxP3$^+$ cells and analysis of IL-10 production on a per-cell basis. CD4$^+$-FoxP3$^+$ Tregs isolated from the spleens of naïve dual reporter mice (*Figure 1—figure supplement 1A and B*) by magnetic bead and sterile fluorescence-activated cell sorting (FACS) were cultured with T cell receptor (TCR) activation using αCD3ε antibody and all combinations of IL-2 and IL-4 for 3 days. We found that Tregs cultured with combinatorial cytokine stimulation resulted in synergistically higher numbers of IL-10 expressing cells (*Figure 1A–C*) and IL-10 secretion (*Figure 1D*) compared to single cytokine stimulation. However, analysis of IL-10$^+$ cells revealed that IL-10 expression as measured by GFP median fluorescence intensity (MFI) was equivalent between IL-2 and IL-2 with IL-4 (*Figure 1E*), suggesting that the cytokines in combination do not elicit a synergistic increase in IL-10 production on a per-cell basis. The sex-independent (*Figure 1F*) and TCR-stimulation-dependent synergy (*Figure 1A–D*) was present in FoxP3$^+$ Tregs but not FoxP3$^-$ Tconv (*Figure 1A*), and no loss of FoxP3 expression was observed (*Figure 1G*), suggesting that the machinery required for this effect was unique to FoxP3$^+$ Tregs. Notably, neither titration of IL-2 concentration from 0.01-fold to 100-fold nor supplementing with αCD28 co-stimulation changed the synergistic and robust effect the combination of IL-2 and IL-4 (IL-2/IL-4) had on IL-10 production by Tregs (*Figure 1—figure supplement 2A and B*).

In order to determine whether the cytokines must be present at the same time, we isolated Tregs and stimulated them with cytokines in series over a 3-day culture and compared the response to simultaneous stimulation. We found that both IL-2 and IL-4 must be present in culture together to manifest the synergistic IL-10 response (*Figure 1H* and *Figure 1—figure supplement 2C*), implicating the integration of IL-2- and IL-4-mediated concurrent intracellular signaling as a key underlying mechanism.

To quantify the effect of IL-2/IL-4 on other Treg-produced cytokines and chemokines, Treg supernatants after 3 days of culture with or without TCR stimulation and all cytokine combinations were analyzed by multianalyte Luminex. Remarkably, no cytokine or chemokine other than IL-10 was both synergistically and robustly increased (*Figure 1—figure supplement 3*) by combined IL-2 and IL-4, including the Treg-associated TGF-β family members (*Tran, 2012*; *Figure 1I*). Moreover, neither IL-2 nor IL-4 induced autocrine production of the reciprocal cytokine (*Figure 1J*).

To differentiate whether the combined cytokines induced or maintained IL-10 expression in Tregs, we sorted CD4$^+$ splenocytes from the dual reporter mice into FoxP3$^+$IL-10$^-$ and FoxP3$^+$IL-10$^+$ populations, then cultured them in all combinations. We observed robust induction of IL-10 expression in FoxP3$^+$IL-10$^-$ cells and maintenance of IL-10 expression in FoxP3$^+$IL-10$^+$ cells, with no detectable change in FoxP3 (*Figure 1K* and *Figure 1—figure supplement 4A*). To assess this effect beyond 3 days, we cultured Tregs and analyzed them on days 3–7 and found a profound daily increase in the overall percentage of IL-10$^+$ Tregs, ultimately leading to nearly every Treg converting to IL-10$^+$ by day 7 (*Figure 1L* and *Figure 1—figure supplement 4B*).

### IL-2/IL-4 enhances Treg proliferation and selectively drives the expansion of IL-10$^+$ Tregs

The lack of a major difference in per-cell IL-10 expression after exposure to IL-2 and IL-4 together (*Figure 1E*) coupled with the conversion of nearly all Tregs into IL-10$^+$ cells (*Figure 1L*) and synergistic IL-10 release (*Figure 1D*) suggested that the cells were proliferating in response to the combined

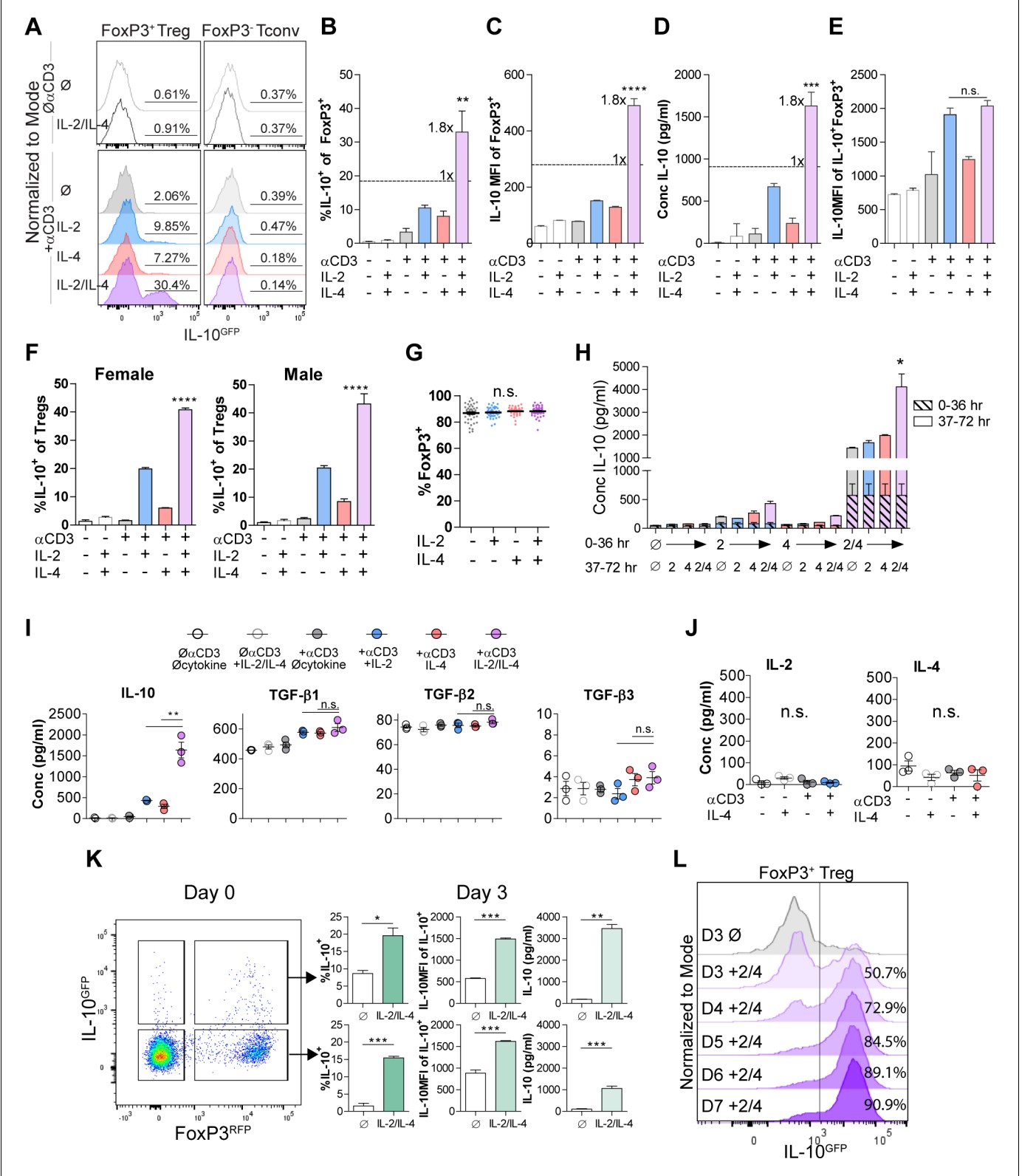

**Figure 1.** IL-2 and IL-4 synergistically promote IL-10 production by Tregs. (**A–C**) IL-10 expression of Tregs purified from *Foxp3*^RFP^/*Il10*^GFP^ dual reporter mice (see also ***Figure 1—figure supplement 1***) cultured for 3 days with the designated stimulants as analyzed by flow cytometry. The IL-2/IL-4 condition is twofold the concentration of the single cytokines. For all panels, N ≥ 3 for all bar graphs and histograms are representative. (**D**) IL-10 production of Tregs cultured with the designated stimulation as quantified by ELISA of the culture supernatants. N = 3. (**E**) IL-10 expression of purified

*Figure 1 continued on next page*

*Figure 1 continued*

IL-10$^+$ Tregs cultured with the designated stimulation as quantified by flow cytometry. N = 3. (F) Female and male responses to combinatorial cytokine stimulation after 3 days, as measured by flow cytometry for IL-10 expression. N = 3. (G) FoxP3 expression by purified Tregs stimulated in culture for 3 days with the designated conditions, as analyzed by flow cytometry. N ≥ 27. (H) IL-10 expression of purified Tregs stimulated for 36 hr in culture, washed, and then subsequently stimulated for another 36 hr in culture with the indicated conditions, as analyzed by flow cytometry. All samples received αCD3ε activation (see also *Figure 1—figure supplement 2C*). N = 3. (I) Cytokine production following 3 days of Treg culture as quantified by multianalyte Luminex of the culture supernatants (see also *Figure 1—figure supplement 3*). N = 3. (J) IL-2 and IL-4 production by Tregs following 3 days of stimulation with the designated conditions, as quantified by ELISA of the culture supernatants. N = 3. (K) IL-10 production of purified IL-10$^+$ or IL-10$^-$ Tregs following 3 days of culture with αCD3ε and combined IL-2/IL-4, as analyzed by flow cytometry (see also *Figure 1—figure supplement 4A*). N = 3. (L) IL-10 production of purified Tregs cultured with αCD3ε and combined IL-2/IL-4 for 3–7 days as analyzed by flow cytometry (see also *Figure 1—figure supplement 4B*). Histograms are representative of three independent experiments. Mean ± SEM are indicated. *p<0.05, **p<0.01, ***p<0.001, ****p<0.0001.

The online version of this article includes the following figure supplement(s) for figure 1:

**Figure supplement 1.** Cell sorting gating strategy and post-sort purity.

**Figure supplement 2.** Dosing and time-dependency of combinatorial cytokine stimulation in Tregs.

**Figure supplement 3.** IL-10 is the only synergistic and robust analyte produced by Tregs following combinatorial cytokine stimulation 31-plex.

**Figure supplement 4.** Dynamics of combinatorial cytokine stimulation of Tregs over time.

cytokines. Dual reporter Tregs were isolated and stained with CellTrace Violet and stimulated with all cytokine combinations for 3 days. Tregs cultured with the combined cytokines were more proliferative, as measured by the number and magnitude of cell division peaks (*Figure 2A*, for gating strategy, see *Figure 2—figure supplement 1A*), proliferation index, and division index (*Figure 2B*). The division index revealed that more Tregs stimulated by the combined cytokines divide, and the proliferation index indicated that each dividing Treg undergoes more divisions than when stimulated with or without the single cytokines. As before, neither co-stimulation with αCD28 (*Figure 2—figure supplement 1B*) nor altering the concentration of IL-2 impacted this effect (*Figure 2—figure supplement 1C*).

Since IL-2 and IL-4 in combination elicited synergistic IL-10 production by Tregs (*Figure 1*), we determined whether Treg proliferation was correlated with higher IL-10 expression among dividing cells. We found that cytokine supplementation led to each subsequent generation of Tregs expressing more IL-10 than the previous generation (*Figure 2C*). Furthermore, staining IL-10$^-$ and IL-10$^+$ Tregs with CellTrace prior to culturing and Sytox viability dye after culturing revealed that originally IL-10$^-$ Tregs that gained IL-10 expression adopted a higher division index than the Tregs which remained IL-10$^-$ (*Figure 2D*), and Tregs that were IL-10$^+$ on day 0 and remained IL-10$^+$ after 3 days of culture had the highest division index and percentage of the overall population that divided. Lastly, the IL-10$^+$ Tregs that lost IL-10 expression had a drastic decrease in proliferation as measured by division index and percentage of cells that underwent division, indicating that IL-10$^+$ Tregs only lose IL-10 expression when they become quiescent and non-dividing.

We then compared the division and proliferation indices of IL-10$^+$ with IL-10$^-$ Tregs and found that IL-10-expressing Tregs were indeed significantly more proliferative than Tregs that do not express IL-10 (*Figure 2E*), and that IL-10$^+$ Tregs supplemented with combined IL-2 and IL-4 proliferated the most of all. Moreover, the increased proliferation and division indices in IL-2/IL-4-treated IL-10$^+$ Tregs compared to IL-10$^-$ Tregs suggests that expression of IL-10 in Tregs, as induced by the combined cytokines, coincides with enhanced proliferation (*Figure 2F*). Using a FoxP3 conditional knockout of IL-10 (IL-10 cKO) created by crossing *Foxp3*$^{cre-YFP}$ (*Rubtsov et al., 2008*) and *Il10*$^{fl/fl}$ (*Roers et al., 2004*) mice, the IL-2/IL-4 enhancement of Treg proliferation was indeed lost (*Figure 2G*), suggesting that IL-10 autocrine signaling may help drive IL-2/IL-4-enhanced Treg proliferation.

Finally, a time course study revealed that of purified Tregs cultured in the presence of all cytokine combinations for 3–7 days, combined IL-2 and IL-4 synergistically increased the number of total Tregs and IL-10$^+$ Tregs over 12-fold in 5 days (*Figure 2H*, *Figure 2—figure supplement 1D*). As seen with IL-10 induction (*Figure 1H*), exposing cells to the cytokines in series rather than together failed to duplicate the proliferative response (*Figure 2—figure supplement 1E*).

Collectively, the data shown in *Figures 1* and *2* demonstrated that not only does the combination of IL-2 and IL-4 lead to strong IL-10 expression and enhanced proliferation among TCR-activated

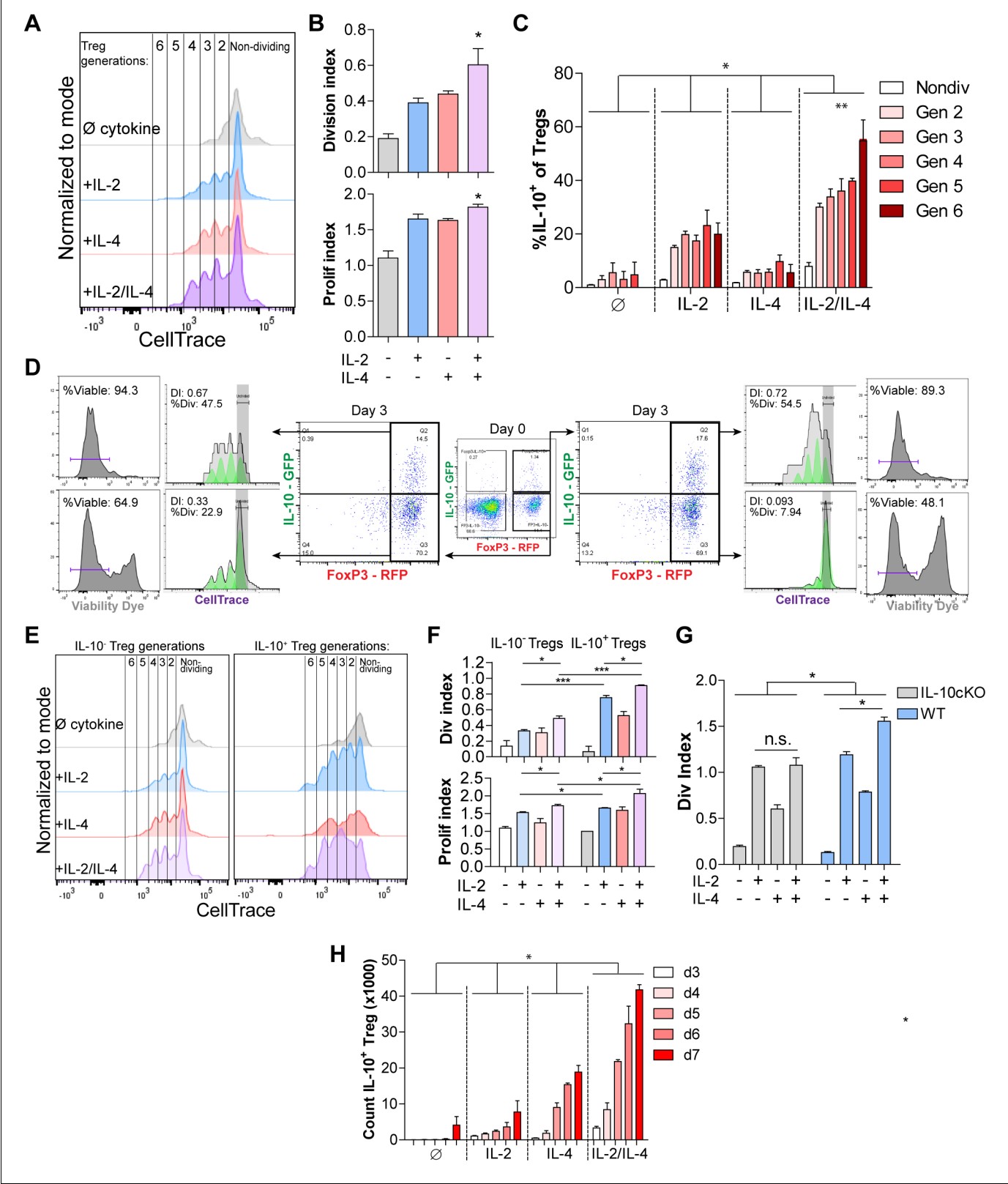

**Figure 2.** IL-2/IL-4 enhances Treg proliferation and selectively drives the expansion of IL-10⁺ Tregs. (**A and B**) Treg proliferation as measured by CellTrace signal following 3 days of culture with αCD3ε and all combinations of IL-2 and IL-4, with division and proliferation indices indicated. N = 3 for all bar graphs and histograms are representative. (**C**) IL-10 expression of purified Tregs cultured for 3 days with αCD3ε and the designated cytokines,

*Figure 2 continued on next page*

*Figure 2 continued*

and gated by CellTrace generation, as measured by flow cytometry (panel **A**). N = 3. The IL-10 expression of all IL-2/IL-4-induced Treg generations are statistically significant (p<0.05) compared to the other cytokine-stimulated conditions. (**D**) Proliferation and viability of purified IL-10$^+$ and IL-10$^-$ Tregs cultured for 3 days with αCD3ε and both IL-2 and IL-4, as flow cytometry analysis of CellTrace and Sytox Red signal. The histograms are representative. N = 3. (**E and F**) Proliferation of purified Tregs cultured for 3 days with αCD3ε and all cytokine combinations, gated on IL-10$^{+/-}$ expression, as analyzed by flow cytometry. N = 3 for all bar graphs and histograms are representative. (**G**) Division index of wild type (FoxP3$^{creYFP}$) and IL-10 cKO (IL-10$^{fl/fl}$× FoxP3$^{creYFP}$) purified Tregs that were stimulated with the indicated conditions for 3 days, as analyzed by flow cytometry. N = 3 for all bars. (**H**) Cell count of IL-10$^+$ Tregs following 3–7 days of culture with αCD3ε and all combinations of IL-2 and IL-4, as analyzed by flow cytometry. N = 3. The cell counts of all IL-2/IL-4-induced conditions are statistically significant (p<0.05) compared to the other cytokine-stimulated conditions. For all panels, mean ± SEM are indicated. *p<0.05, **p<0.01, ***p<0.001.

The online version of this article includes the following figure supplement(s) for figure 2:

**Figure supplement 1.** Dosing and time-dependency of combinatorial cytokine stimulation on Treg proliferation.

Tregs, but that these events were associated in such a way that led to the selective propagation of IL-10$^+$ Tregs resulting in an exponentially increased Treg response.

## Combined IL-2 and IL-4 increases the suppressive ability of Tregs

The ability of IL-10 to suppress the proliferation and functions of conventional and effector T cells is well documented (*Akdis and Blaser, 2001*). Since the cytokines working together elicit extraordinarily robust IL-10 production by Tregs through a combination of IL-10 induction and cellular proliferation, we assessed the in vitro suppressive impact on Tconv proliferation and cytokine production. Tregs were stimulated as before for 3 days, washed, counted, and normalized to the number of surviving Tregs, and placed them in co-culture with freshly purified and CellTrace-stained Tconv cells harvested from the spleens of dual reporter mice. After 3 days, the cells were analyzed for proliferation and cytokine production. Treg suppression activity was impaired upon IL-4 stimulation, despite the IL-4-mediated increase in IL-10 (*Figure 1*), while supplementation with either IL-2 or both cytokines substantially increased Tconv inhibition (*Figure 3A*). Stimulation by combined IL-2 and IL-4 led to increased IL-10 release by Tregs (*Figure 3B*, *Figure 3—source data 1A*) and significant suppression of IL-4 and IL-17A production by Tconv cells (*Figure 3C*, *Figure 3—source data 1A*), but only a modest increase in suppressive capacity over IL-2 exposure alone.

Next, to assess the role of IL-10 in Tconv inhibition, we repeated the experiment using Tregs isolated from *Il10*-deficient (IL-10$^{-/-}$) mice and found that cytokine-stimulated IL-10$^{-/-}$ Tregs failed to suppress the proliferation of WT Tconv cells (*Figure 3D*), did not express IL-10 (*Figure 3E*, *Figure 3—source data 1B*), and failed to reduce Tconv cytokine expression (*Figure 3F*, *Figure 3—source data 1B*). Likewise, Treg-specific IL-10 cKO cells also failed to inhibit WT Tconv proliferation and cytokine production, due to a lack of IL-10 release (*Figure 3G–I*, *Figure 3—source data 1C*). Consistent with the known uneven disease penetrance in mice lacking IL-10 (Jax), neither the IL-10$^{-/-}$ nor IL-10 cKO mice used for these experiments had developed colitis (*Figure 3—figure supplement 1A and B*), thereby reducing confounding factors associated with T cell function. These data indicated that on a per-cell basis, Tregs stimulated with IL-2 alone had similar suppressive activity compared to the IL-2/IL-4 combination, and that this activity was IL-10 dependent. This agrees with the lack of difference in per-cell IL-10 production (*Figure 1E*), but failed to account for the enhanced and selective proliferation of IL-10$^+$ Tregs with combinatorial cytokine exposure.

To explore the impact of the Treg proliferative response, we cultured Tregs alone with all combinations of cytokines for 7 days (*Figure 3A–C*) and washed them after stimulation, but did not re-normalize the cell numbers prior to co-culture with fresh Tconv cells to allow for proliferative differences to be incorporated into the activity assessment. We observed that Tregs cultured without cytokine had lost their suppressive ability, likely due to an extended amount of time without supportive growth signals in culture, and that the Tregs cultured with single cytokines maintained suppressive activity (*Figure 3J–L*, *Figure 3—source data 1D*). However, Tregs stimulated with both cytokines showed significantly greater suppression of Tconv proliferation (*Figure 3J*), produced higher concentrations of IL-10 (*Figure 3K*), and suppressed the production of IL-4 and IL-17A by Tconv cells to a greater degree (*Figure 3L*) compared to single cytokine conditions. Indeed, we found that by day 9 of a 1:8 Treg-Tconv co-culture, the WT Tregs previously stimulated with both IL-2 and IL-4 overtook greater than 80% of the co-culture whereas IL-10-deficient Tregs from IL-10 cKO mice did not

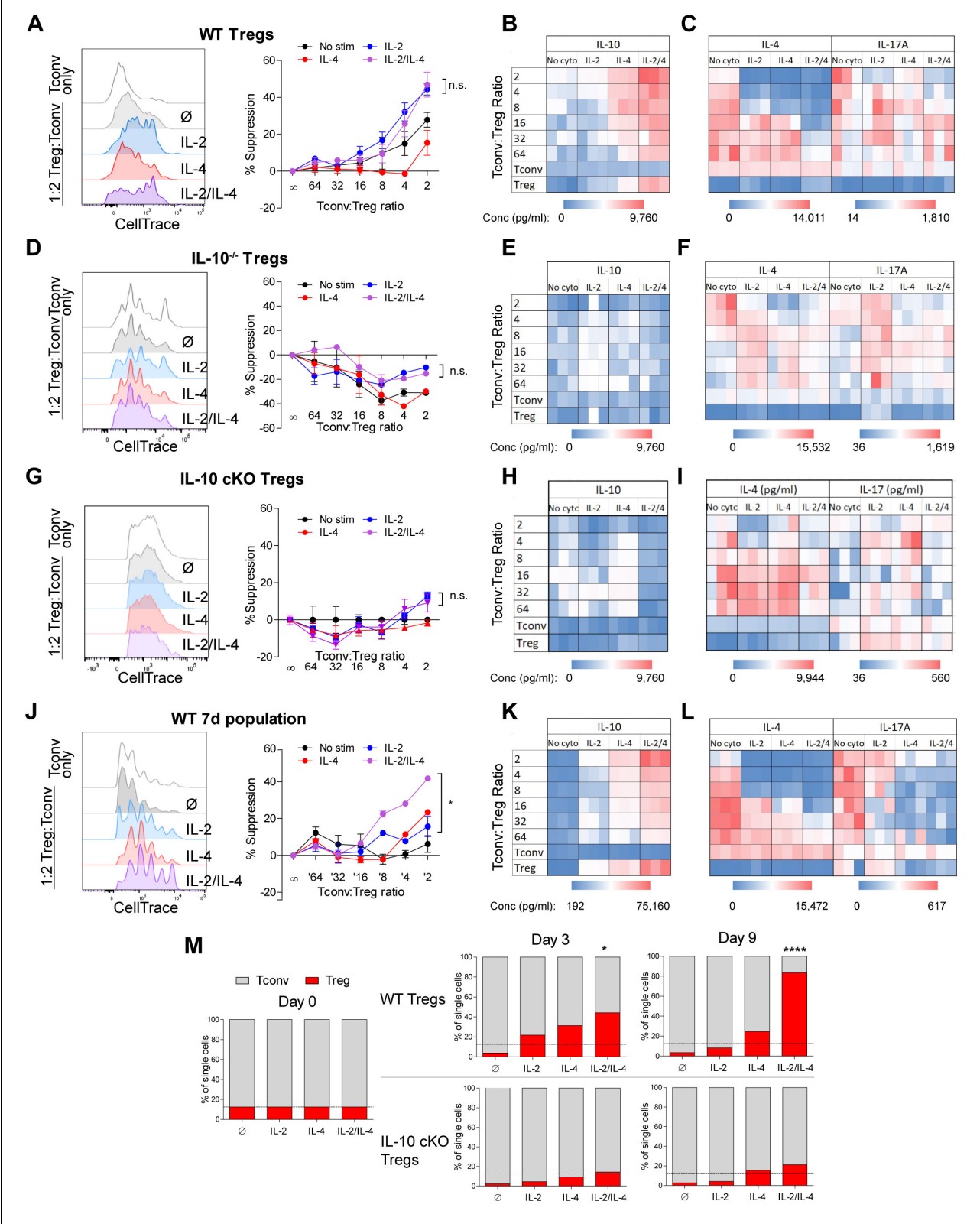

**Figure 3.** Combined IL-2 and IL-4 increase the suppressive ability of Tregs. (A–C) Proliferation (flow cytometry) and cytokine output (ELISA) of freshly isolated and Celltrace-stained Tconv co-cultured with αCD3ε and the indicated ratios of WT Tregs separately and previously stimulated with αCD3ε and all cytokine conditions for 3 days, washed, and normalized for cell number at the time of co-culture. The flow histograms were gated on Tconv cells. N = 3 for all graphs and histograms are representative. (D–F) Proliferation (flow cytometry) and cytokine production (ELISA) of freshly isolated

*Figure 3 continued on next page*

*Figure 3 continued*

Tconv cells co-cultured with αCD3ε and the indicated ratios of IL-10$^{-/-}$ Tregs separately and previously stimulated with αCD3ε and all cytokine conditions for 3 days, washed, and normalized for cell number at the time of co-culture. The flow histograms were gated on Tconv cells. N = 3 for all graphs and histograms are representative. (G–I) Proliferation (flow cytometry) and cytokine production (ELISA) of freshly isolated Tconv cells from WT mice co-cultured with αCD3ε and the indicated ratios of IL-10 cKO Tregs separately and previously stimulated with αCD3ε and all cytokine conditions for 3 days, then washed. The flow histograms were gated on Tconv cells. N = 3 for all graphs and histograms are representative. (J–L) Proliferation (flow cytometry) and cytokine production (ELISA) of freshly isolated Tconv cells co-cultured with αCD3ε and the indicated ratios of WT Tregs separately and previously stimulated with αCD3ε and all cytokine conditions for 7 days. Co-culture ratio was based on the number of Tregs prior to cytokine stimulation to incorporate their proliferation. The flow histograms were gated on Tconv cells. N = 3 for all graphs and histograms are representative. (M) Percentage of the overall culture comprising of WT or IL-10 cKO Tregs and Tconv over the course of time as determined by flow cytometry. The Tregs were purified and stimulated with αCD3ε and all cytokine conditions for 3 days, washed, then placed in co-culture with freshly isolated Tconv cells at a 1:8 Treg:Tconv starting ratio (line). N = 3. For all panels, mean ± SEM are indicated. *p<0.05, ****p<0.0001.

The online version of this article includes the following source data and figure supplement(s) for figure 3:

**Source data 1.** Numeric values of the heatmaps in *Figure 3B–C* (A), *Figure 3E–F* (B), *Figure 3H–I* (C), and *Figure 3K–L* (D).
**Figure supplement 1.** Colon lengths of conditional and global Il10 knockout mice.

(*Figure 3M*). These demonstrated that the synergistic impact of these cytokines was manifested through the combination of IL-10 production, suppression of Tconv proliferation and cytokine production, and the selective proliferation of IL-10$^+$ Tregs even after removal from recombinant cytokine stimulation.

## Synergistic IL-10 production and proliferation is STAT5-dependent

The intracellular signaling downstream of IL-2 and IL-4 receptors is well known, including STAT6 as characteristic of IL-4 signaling (*Takeda et al., 1996*) and STAT5 being associated with IL-10 production in Tregs downstream of IL-2 (41). In order to understand the mechanism by which IL-2 and IL-4 signals are integrated, we measured STAT phosphorylation by flow cytometry following Treg stimulation as before. While STAT3 is associated with both IL-2 and IL-4, we found that STAT3 phosphorylation was not altered as a result of combined cytokine stimulation in TCR-activated Tregs, although a small amount of STAT3 phosphorylation occurred at 60 min with combined cytokines without TCR stimulation (*Figure 4A*). IL-2 induced low levels of STAT6 phosphorylation, while IL-4 induced robust STAT6 phosphorylation in TCR-activated Tregs. Remarkably, the addition of IL-2 to the IL-4 and TCR signals resulted in the reversal of IL-4-dependent STAT6 activation (*Figure 4B*).

We also found that STAT5 phosphorylation was weak downstream of either independent cytokine at the concentrations used herein (*Figure 4C*), although consistent with prior reports (*Tsuji-Takayama et al., 2008*), 100-fold higher concentrations of IL-2 was able to induce STAT5 activation (*Figure 4—figure supplement 1A*). In combination, however, IL-2 and IL-4 at low concentrations led to a dramatic and synergistic increase in STAT5 phosphorylation, suggesting that STAT5 may be necessary for the synergistic IL-10 and proliferative response in Tregs.

In order to determine whether STAT5 activation is a point of signaling convergence necessary for synergy, we blocked STAT5 activity over a range of increasing inhibitor (STAT5i) doses in stimulated Tregs. Blockade of STAT5 activity inhibited the response by over 80% in a dose-dependent fashion, as measured by inhibition of IL-10 induction as a percent (*Figure 4D*) and number (*Figure 4E*) of IL-10$^+$ Tregs, overall Treg viability (*Figure 4—figure supplement 1B*), IL-10 expression on a cellular level (*Figure 4—figure supplement 1C*), proliferation (*Figure 4—figure supplement 1D and E*), and IL-10 release (*Figure 4—figure supplement 1F*). As a control, analysis of IFNγ, a cytokine not produced synergistically following combined IL-2 and IL-4 stimulation demonstrated no inhibition (*Figure 4—figure supplement 1G*).

These data indicated that IL-10 production and proliferation downstream of combined IL-2 and IL-4 was dependent upon STAT5; however, it remained unclear whether cytokine synergy also required STAT5. We again cultured Tregs with the single or combinatorial cytokines and a range of increasing concentrations of STAT5i. Although STAT5i inhibited IL-10 expression and reduced the number of IL-10$^+$ Tregs under all conditions (*Figure 4F and G*), we discovered that at high STAT5i concentration, IL-2/IL-4 synergy was lost, as illustrated by a lack of difference between the sum of the individual IL-2 and IL-4 responses (*Figure 4F and G*; stacked bars) and stimulation with both cytokines simultaneously (*Figure 4F and G*; purple bars).

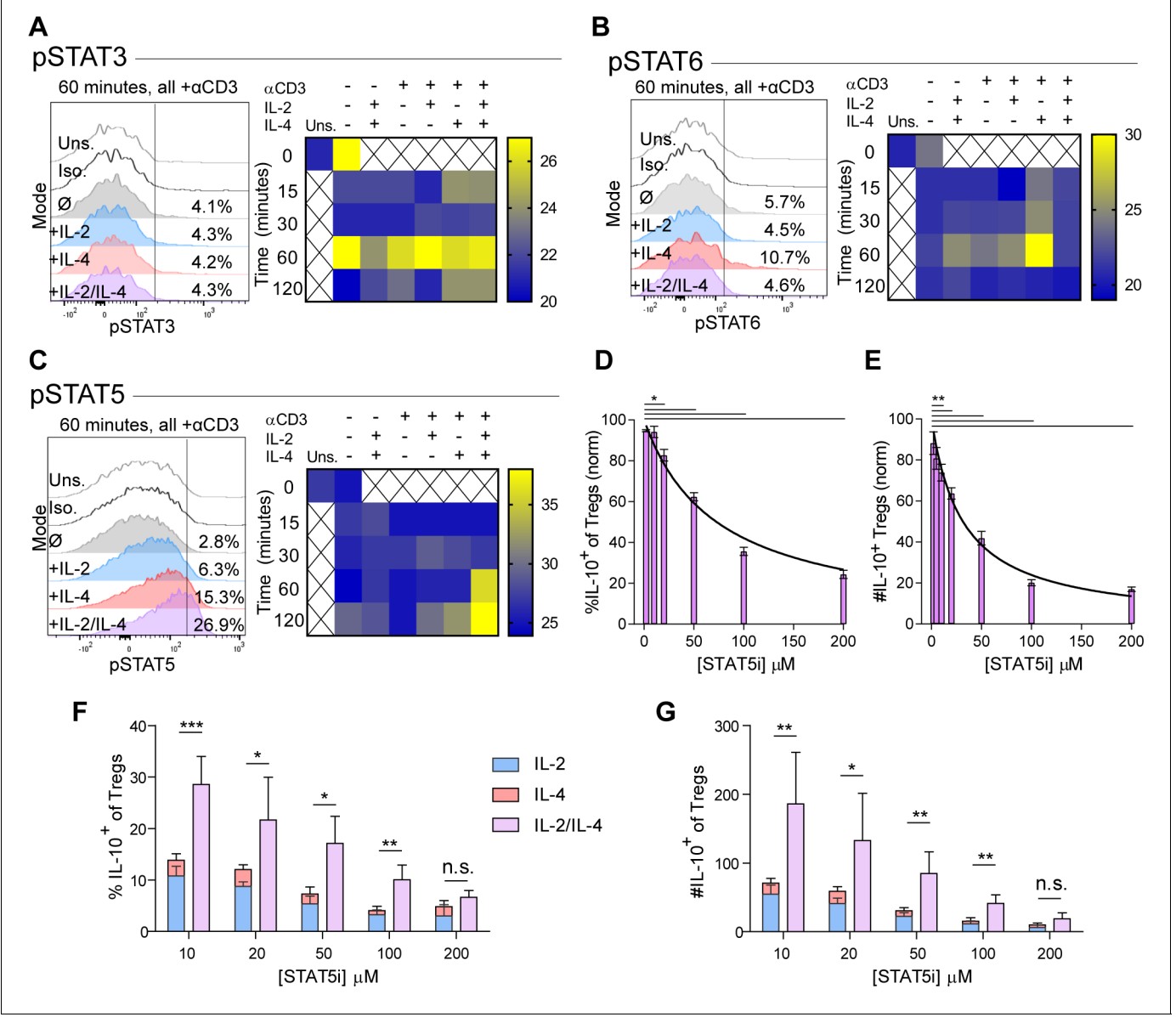

**Figure 4.** Synergistic IL-10 production and proliferation is STAT5-dependent. (**A–C**) pSTAT3, pSTAT6, and pSTAT5 expression of purified Tregs stimulated with the indicated culture conditions as analyzed by flow cytometry. The histograms are representative and heatmaps of median fluorescence intensity (MFI) represent N ≥ 3 experiments. (**D and E**) Treg expression of IL-10 and enumeration of IL-10+ Tregs following αCD3ε, combined IL-2/IL-4, and STAT5i supplementation in 3-day culture, as analyzed by flow cytometry (see also *Figure 4—figure supplement 1*). N = 3. (**F and G**) IL-10 expression in Tregs and IL-10+ Tregs counts following αCD3ε, all combinations of IL-2 and IL-4, and STAT5i supplementation in 3-day culture, as analyzed by flow cytometry. N = 3. For all panels, mean ± SEM are indicated. *p<0.05, **p<0.01, ***p<0.001.
The online version of this article includes the following figure supplement(s) for figure 4:

**Figure supplement 1.** The IL-10 and proliferative response following combined IL-2 and IL-4 stimulation is dependent on STAT5 signaling.

These data demonstrated that the signaling downstream of individual cytokine receptors was altered by the combinatorial integration of other cytokine-stimulated signals, as was clear for STAT5 and STAT6 activation. Moreover, not only was IL-10 production and Treg proliferation paired events, but STAT5 was necessary to induce the synergistic response characteristic of the IL-2 and IL-4 combination.

## Combinatorial IL-2 and IL-4 signaling promotes the expression of the type 2 IL-4 receptor

We have found that IL-2 and IL-4 in combination promoted selective proliferation of IL-10$^+$ Tregs (*Figure 2*), suggesting that cytokine receptor expression may change over time to promote the maintenance of IL-10 expression. Thus, we measured cell surface expression of all IL-2, IL-4, and IL-10 receptor subunits. Tregs were again stimulated, and each subunit was quantified by flow cytometry. Surface expression was first visualized using principle component analysis (PCA), which demonstrated that the receptor expression pattern of TCR-activated Tregs without cytokines, with IL-2 or IL-4 individually, and both cytokines were all divergent and distinct (*Figure 5A*), further supporting a model in which Tregs possess the ability to integrate multiple cytokine signals in ways distinct from simply adding known pathways studied in isolation.

Further analyses of the data revealed a strong positive correlation between IL-10 and IL-2Rα (CD25) expression (*Figure 5B*), which agrees with both a prior study reporting STAT5-dependent expression of IL-2Rα on Tregs (*Kim et al., 2001*) and our STAT5 data (*Figure 4C*). We also discovered a strong negative correlation between IL-10 and IL-2Rβ expression (*Figure 5C*), and although there was a positive correlation between IL-10 and γC chain in cells receiving individual cytokines, this was significantly reduced when IL-2 and IL-4 were together (*Figure 5D*). Looking at the IL-4 receptor subunits, the IL-4Rα expression pattern (*Figure 5E*) mirrored that observed for the γC chain (*Figure 5D*), with a positive correlation with IL-10 for the individual cytokines, but less so with the combination. Curiously, IL-13Rα1 expression showed a strong positive correlation with IL-10 (*Figure 5F*), similar to IL-2Rα (*Figure 5B*), following combined cytokine stimulation. Since the type II IL-4R is used by both IL-4 and IL-13, we tested whether IL-13 could achieve the same synergistic Treg outcome as IL-4 in combination with IL-2. Supplementation of Treg cultures with recombinant IL-13 in any combination with IL-2 and IL-4 did not impact IL-10 production (*Figure 5G and H*), suggesting that the response is specific for IL-4. Together, these data demonstrated that IL-2Rα and the IL-4-mediated stimulation of the type II IL-4R (IL-4Rα + IL-13Rα) were selectively associated with the synergistic response to the IL-2 and IL-4 combination.

Analysis of the IL-10R (*Figure 5I*) further revealed a similar pattern as observed for the γC chain (*Figure 5D*), with a reduction in expression after exposure to both cytokines compared to IL-2 alone. In order to determine whether IL-10-IL-10R autocrine signaling may also be playing a role in synergistic IL-10 production (*Figure 1*), we blocked the IL-10R via neutralizing antibody at two concentrations. We found a profound loss of IL-10 expression downstream of IL-2/IL-4 on a per-cell basis (*Figure 5J*), but only a modest decrease in proliferation using high concentrations of the IL-10R antibody compared to isotype control (*Figure 5—figure supplement 1*). These findings indicate that IL-10 signaling promoted further IL-10 expression in Tregs stimulated with IL-2 and IL-4, but only played a minor role in their proliferation.

## Combinatorial IL-2 and IL-4 signaling suppresses asthma-like pulmonary morbidity

Given the potent in vitro Treg response to the combination of IL-2 and IL-4 and the resulting integration of their signaling pathways, we sought to determine the in vivo response within the context of inflammatory disease. Since this pathway was initially discovered in murine asthma (*Jones et al., 2019*; *Johnson et al., 2018*; *Johnson et al., 2015a*; *Johnson et al., 2015b*), we first challenged mice with house dust mite (HDM) over the course of 2 weeks to induce pulmonary inflammation as described previously (*Jones et al., 2019*). The combinatorial cytokines were administered i.n. 1 week following the first HDM challenge (*Figure 6—figure supplement 1A*). We found that a 1:1 ratio of IL-2/IL-4 was sufficient to suppress disease, as indicated by reduced bronchoalveolar lavage (BAL) cell differentials (*Figure 6A*) and greatly improved lung histology (*Figure 6B*). In comparison, individual cytokines administered at the same doses failed to significantly suppress infiltration (*Figure 6A*) or tissue pathology (*Figure 6B*). Flow cytometric analysis of lung and spleen cells showed that the mice administered combined cytokines accumulated more IL-10$^+$ and total Tregs per gram of tissue (*Figure 6C*), and that IL-10 expression in FoxP3$^-$ Tconv cells and CD4$^-$ cells did not increase (*Figure 6—figure supplement 1B and C*). Unlike in WT mice in both Th1-skewed C57Bl/6 and Th2-skewed BALB/c backgrounds, IL-2/IL-4 failed to reduce disease morbidity in IL-10 cKO mice (C57Bl/6 background), as measured by BAL cell differentials (*Figure 6D*), H and E tissue

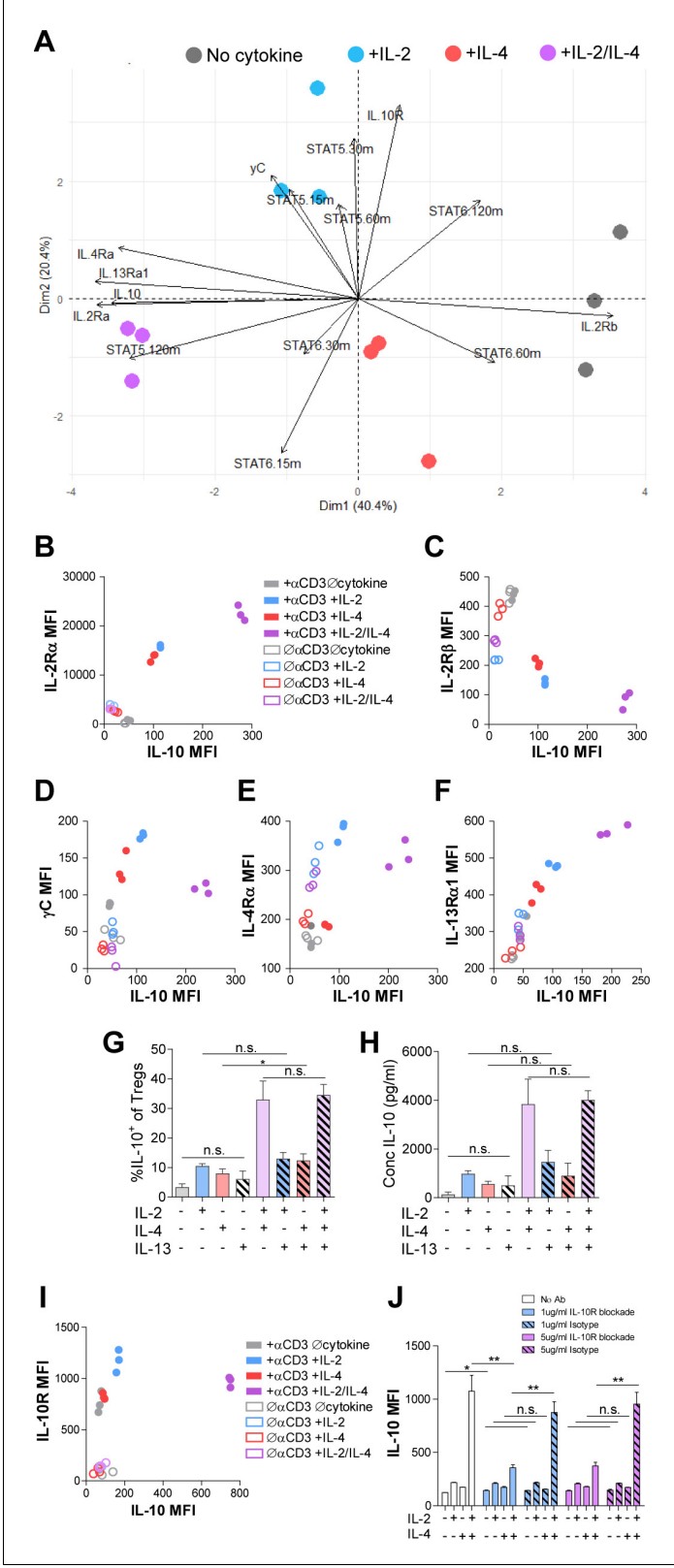

**Figure 5.** Combinatorial IL-2 and IL-4 signaling promotes the expression of the Type 2 IL-4 receptor. (**A**) Expression of 14 parameters as shown in an unpaired principle component analysis of Tregs purified and stimulated with αCD3ε and all cytokine conditions for 3 days unless otherwise noted in the vectors. N = 3. (**B–F**) Surface expression of receptor subunits that comprise the multiple forms of the IL-2R and IL-4 on purified Tregs

*Figure 5 continued on next page*

*Figure 5 continued*

from $Foxp3^{RFP}/Il10^{GFP}$ dual reporter mice that were stimulated for 3 days with or without αCD3ε and with all cytokine combinations. Expression of receptor subunits is plotted as a function of IL-10 expression. N = 3. (G and H) Quantification of IL-10 expression and protein concentration secreted by Tregs that were purified and cultured for 3 days with αCD3ε and all combinations of IL-2, IL-4, and IL-13, as analyzed by flow cytometry and ELISA, respectively. N = 3. (I) Surface expression of IL-10R on purified Tregs from $Foxp3^{RFP}/Il10^{GFP}$ dual reporter mice that were stimulated for 3 days with or without αCD3ε and with all cytokine combinations. Expression of receptor subunits is plotted as a function of IL-10 expression. N = 3. (J) IL-10 expression of purified Tregs stimulated with αCD3ε and all cytokine conditions for 3 days with or without neutralizing IL-10R monoclonal antibody or isotype control blockade, as analyzed by flow cytometry (see also *Figure 5—figure supplement 1*). N = 3. For all panels, mean ± SEM are indicated. *p<0.05.

The online version of this article includes the following figure supplement(s) for figure 5:

**Figure supplement 1.** IL-10R blockade modestly reduces IL-2/IL-4-induced Treg proliferation.

histology of infiltrating cells and bronchial epithelial cell hyperplasia (*Figure 6E* and *Figure 6—figure supplement 1D*), and PAS staining of mucus production (*Figure 6F*). In fact, mucus production was increased in IL-2/IL-4-treated asthmatic IL-10 cKO mice (*Figure 6F*).

We further investigated the efficacy of the combined cytokines administered prior to disease induction as prophylactic/preventative therapy rather than a treatment of established disease (*Figure 6—figure supplement 2A*). We found that mice pretreated with IL-2/IL-4 were significantly protected from developing airway disease as quantified by BAL differentials and tissue pathology (*Figure 6G*, *Figure 6—figure supplement 2B*), but that mice deficient in IL-10 were not protected and showed increased severity of disease overall (*Figure 6G*, *Figure 6—figure supplement 2C*).

Finally, asthma in human patients often persists as chronic disease. We therefore tested the efficacy of the cytokines in a chronic model of pulmonary inflammation induced by challenging mice with three alternating antigens (i.e. HDM, cockroach antigen, and ovalbumin) over 10 weeks (*Figure 6—figure supplement 3A*). Since this model results in tissue remodeling and hyper-responsiveness to bronchial restricting agents, we examined the lung function of the mice using a Flexivent ventilator, and observed that the mice treated with the combined cytokines had significantly lower lung resistance upon methacholine challenge (*Figure 6H*). Moreover, lung histology revealed reduced cellular infiltration and epithelial hyperplasia by H and E staining, decreased mucus production (PAS staining), and prevention of long-term tissue remodeling (TriChrome staining) when provided IL-2 and IL-4 combination therapy (*Figure 6—figure supplement 3B*).

These data revealed that the combination of IL-2 and IL-4 both prevented and reversed pulmonary inflammatory disease in multiple murine models on both Th1 and Th2-skewed backgrounds.

## Combinatorial IL-2 and IL-4 signaling suppresses EAE morbidity

Since the asthma models allow for direct administration of cytokine to the site of inflammation via inhalation, we sought to determine whether systemic administration of both cytokines would also be beneficial in IL-17-driven EAE model. EAE was induced in dual reporter mice by sensitization with myelin oligodendrocyte glycoprotein (MOG)$_{35-55}$ peptide and adjuvant. We found that a 1:1 ratio of IL-2/IL-4 was sufficient to suppress disease, as indicated by reduced clinical score (out of 5) (*Figure 7A*, *Figure 7—figure supplement 1A*), as well as reduced demyelination and cellular infiltration on spinal cord histology (*Figure 7B*). In comparison, individual cytokines administered at the same doses failed to suppress disease. Importantly, both IL-2 and IL-4 have individually shown efficacy for EAE suppression in previous reports; however, the IL-2 and IL-4 concentrations were 36-fold (*Rouse et al., 2013*) and 145-fold (*Racke et al., 1994*) greater, respectively, in those prior studies compared to the concentrations used herein.

To examine the efficacy of the combined cytokine therapy at different stages of disease development, the cytokines were injected i.v. into mice challenged with MOG. All three delivery schedules (*Figure 7—figure supplement 2A*), including preventative (i.e. prophylactic), concomitant (i.e. at the first sign of disease), and therapeutic (i.e. approaching peak disease), generated significant reductions in clinical score (*Figure 7C* and *Figure 7—figure supplement 2A–C*). Histological analyses indicated that mice treated with the combined cytokines had reduced cellular infiltration (*Figure 7—figure supplement 2C*; H and E) and demyelination (*Figure 7—figure supplement 2C*; H and E;

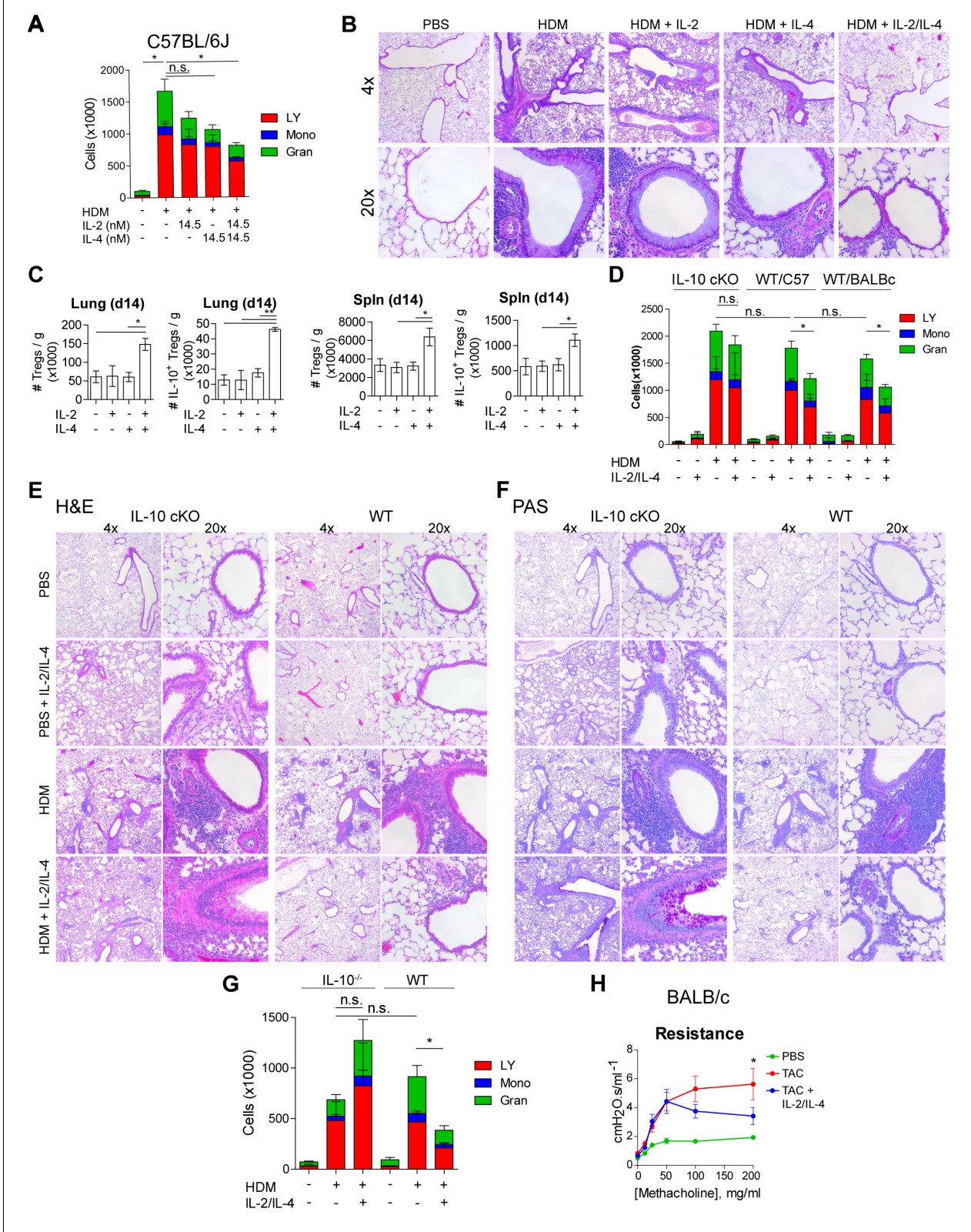

**Figure 6.** IL-2 and IL-4 in combination suppress the severity of HDM-induced asthma. (**A and B**) Analysis of HDM-induced airway inflammation in WT C57Bl/6J mice by cell differentials of the Bronchial Alveolar Lavage (BAL, LY = lymphocytes, Mono = monocytes, Gran = granulocytes) and hematoxylin and eosin (H&E) staining of FFPE-processed pulmonary tissues. IL-2 and IL-4 were administered i.n. on days 7–11 of the acute therapeutic trial (see also A). N = 6. Histological images are representative. (**C**) Quantification of Treg and IL-10[+] Treg numbers in naïve mice following i.n. administration of

*Figure 6 continued on next page*

Figure 6 continued

combined IL-2 and IL-4 on days 0–5. Mice were harvested at day 14 for flow cytometric analysis. N = 6. (D–F) Assessment of HDM-induced airway inflammation in IL-10 cKO C57Bl/6J mice, WT C57Bl/6J mice, and WT BALB/c mice by cell differential analysis of the BAL and H and E and PAS staining of FFP3-processed pulmonary tissues. Mice received i.n. challenge with HDM and i.n. administration of combined IL-2 and IL-4 in an acute therapeutic regimen on days 7–11 (see also A–D). N = 6. (G) Analysis of HDM-induced airway inflammation in *Il10*[-/-] C57Bl/6J mice and WT C57Bl/6J mice by cell differentials of the Bronchial Alveolar Lavage fluid. IL-2 and IL-4 were administered i.n. on days 0–5 according to the acute preventative trial regimen (see also *Figure 6—figure supplement 2*). N = 6. (H) Airway resistance in chronic triple antigen-challenged (TAC; HDM/CRA/OVA) BALB/c mice treated or not with i.n. IL-2 and IL-4 starting in week 3 (see also *Figure 6—figure supplement 3*). Airway resistance was measured by Flexivent as a dose response to methacholine. N ≥ 2. For all panels, mean ± SEM are indicated. *p<0.05, **p<0.01, ***p<0.001.

The online version of this article includes the following figure supplement(s) for figure 6:

**Figure supplement 1.** Administration of IL-2 and IL-4 suppresses HDM-induced acute asthma and is dependent on IL-10.

**Figure supplement 2.** Administration of prophylactic IL-2/IL-4 suppresses HDM-induced acute asthma.

**Figure supplement 3.** Administration of IL-2 and IL-4 therapeutically suppresses TAC-induced chronic asthma.

Luxol Fast Blue) of the spinal cord. Flow cytometry further revealed that the cytokines led to significantly reduced MOG-specific Tconv cells (*Figure 7D*) while ELISA showed decreased IL-17 within neuronal tissues (*Figure 7E*).

To better understand the underlying mechanism of EAE suppression, we analyzed Tregs within the central nervous system (CNS) and spleen. We found that IL-2 and IL-4 in combination increased the percentage of CNS and splenic IL-10[+] Tregs in MOG-challenged animals (*Figure 7F and G*) in correlation with reduced disease. Interestingly, we did not observe increases in IL-10 among CNS-localized Tconv cells (*Figure 7—figure supplement 3A*) or F4/80[+] macrophages in MOG-challenged mice with cytokine therapy (*Figure 7H*) despite IL-4 being a classical M2-polarizing cytokine in macrophages (*Loke et al., 2002*).

Finally, we found that cytokine efficacy in EAE was lost in both germline IL-10[-/-] and Treg-specific IL-10 cKO mice, as measured by disease score (*Figure 7I–K*, *Figure 7—figure supplement 3B*), cellular infiltration and demyelination (*Figure 7J*, *Figure 7—figure supplement 3B and C*), and IL-17A levels in the CNS (*Figure 7L*). These findings collectively indicated that IL-2/IL-4 combination therapy reduced EAE disease burden through Treg-specific release of IL-10.

## Discussion

In this study, we discovered that the combinatorial signaling mediated by IL-2 and IL-4 in TCR-activated Tregs produced a response characterized by IL-10 expression synergy, selective proliferation of IL-10[+] Tregs, differential STAT5 and STAT6 signaling, and distinct cell surface receptor phenotype compared to either cytokine in isolation. These factors cumulatively enhanced the ability of Tregs to suppress Tconv cell proliferation and cytokine secretion in co-culture while successfully attenuating inflammatory diseases of different underlying etiologies in preventative, concomitant, and therapeutic delivery regimens. Disease suppression was paired with increased Treg prevalence and IL-10 expression in Tregs, a reduction in MOG-specific effector T cells in EAE, and was abrogated in mice deficient in IL-10, both globally and conditionally in FoxP3[+] Tregs. Together, our data reveal that the combinatorial cytokine signaling in Tregs is integrated to generate a unique biological outcome that is divergent from the sum of each pathway in isolation and robustly enhances and maintains the suppressive ability of FoxP3[+] Tregs in vitro and in vivo.

Recently, the global COVID-19 pandemic has brought attention to the devastating effects of cytokine storms, with one study finding that TNFα and IFNγ synergize to trigger inflammatory cell death and increased mortality in SARS-CoV-2 infection (*Karki et al., 2021*). Mechanistically, previous studies have shown that two cytokines can cooperate by sharing the same STAT proteins to either enhance or interfere with the other to form a different outcome (*Lin and Leonard, 2019*; *Johnston et al., 2012*; *Liao et al., 2011*), thereby demonstrating the importance of understanding how cytokines may influence each other. While our study focuses on Tregs, others have reported optimal differentiation of Th2 cells with sequential IL-2 and then IL-4 stimulation, which upregulates IL-4Rα and is mediated by combined STAT5 and STAT6 phosphorylation (*Cote-Sierra et al., 2004*; *Zhu et al., 2003*). Compared to sequential nature of these and other studies (*Lin and Leonard, 2019*), our principal findings required simultaneous stimulation by both cytokines (*Figure 1H*).

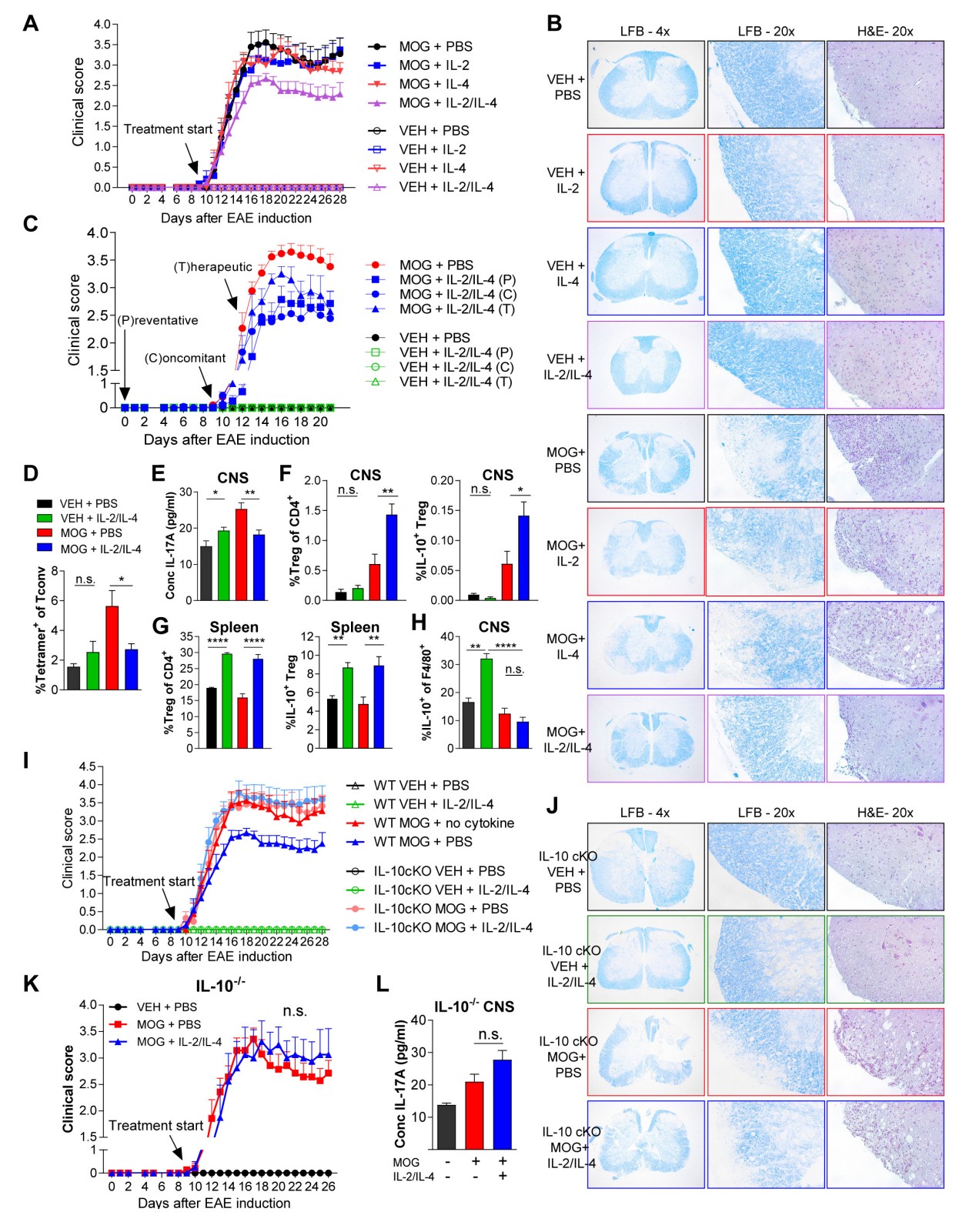

**Figure 7.** Combined IL-2 and IL-4 reduces the severity of experimental autoimmune encephalomyelitis (EAE). (**A and B**) EAE clinical score and FFPE-processed spinal cords sectioned and stained with H and E and LFB from mice challenged with myelin oligodendrocyte glycoprotein (MOG) peptide with and without combined IL-2/IL-4 and subclinically effective doses of the individual cytokines (For statistical analyses, see *Figure 7—figure supplement 1*). Histological images of H and E and Luxol Fast Blue (LFB) staining represent a mouse of the mean disease score of each condition. *Figure 7 continued on next page*

*Figure 7 continued*

N = 10–12 for EAE mice; N = 4 for vehicle control mice. (**C**) EAE clinical score and FFPE-processed spinal cords sectioned and stained with H and E and LFB from mice challenged with MOG peptide with and without the combined cytokines according to the indicated dosing schedule (for statistical analyses, see *Figure 7—figure supplement 2A and B*). N = 10–12 for EAE mice; N = 4 for vehicle control mice. (**D**) Analysis of MOG-loaded MHCII tetramer positive Tconv cells in the CNS as determined by flow cytometry. The color legend is indicated for panels **D–H**. N = 3 (**E**) Quantification of IL-17A secreted by CNS leukocytes isolated by Percoll gradient and incubated overnight in media, as quantified by ELISA. The conditions are described by the key in 7C. N = 3. (**F and G**) Analysis of Treg abundance and IL-10-expression in Tregs isolated from the CNS or spleen of *Foxp3^RFP^/Il10^GFP^* mice in the EAE trials by flow cytometry. The conditions are described by the key in 7C. N = 3. (**H**) Analysis of IL-10 expression of F4/80^+^ CNS macrophages from *Foxp3^RFP^/Il10^GFP^* mice in the EAE trials by flow cytometry. The conditions are described by the key in 7C. N = 3. (**I and J**) Assessment of EAE severity in IL-10 cKO mice quantified by daily clinical scoring (For statistical analyses, see *Figure 7—figure supplement 3B*) and FFPE-processing of spinal cords stained with H and E and LFB. Images represent a mouse of the mean disease score of each condition. N = 10–12 for EAE mice; N = 4 for negative control mice for disease. N = 10–12 for EAE mice; N = 4 for vehicle control mice. (**K**) Assessment of EAE severity in *Il10^-/-^* mice as quantified by daily clinical scoring. N = 7–8 for EAE mice; N = 4 for negative control mice for disease (see also *Figure 7—figure supplement 3C*). (**L**) Quantification of IL-17A secreted by CNS leukocytes of *Il10^-/-^* mice isolated by Percoll gradient and incubated overnight in media, as quantified by ELISA. N = 3. For all panels, mean ± SEM are indicated. *p<0.05, **p<0.01, ****p<0.0001.

The online version of this article includes the following figure supplement(s) for figure 7:

**Figure supplement 1.** Combined IL-2 and IL-4 suppressed the severity of experimental autoimmune encephalomyelitis (EAE).
**Figure supplement 2.** Combined IL-2 and IL-4 suppressed the severity of experimental autoimmune encephalomyelitis (EAE) in preventative, concomitant, and therapeutic dosing regiment.
**Figure supplement 3.** IL-10 is critical for suppression of experimental autoimmune encephalomyelitis (EAE) by IL-2/IL-4.

Furthermore, while FoxP3^-^ Tconv cells undergo a Th2-skewing response to IL-4 and IL-2 (52), we found that this combination failed to induce IL-10 production in FoxP3^-^ Tconv cells (*Figure 1A*) while programming FoxP3^+^ Tregs toward an IL-10^+^ and suppressive phenotype. Therefore, our data reveal that not only can the integration of multiple cytokine pathways diverge from the sum of the parts, the combinatorial nature is further complicated by an enormous variety of responding cell types and disease states.

Interestingly, our data indicate that simultaneous administration of IL-2 and IL-4 in vivo suppressed the severity of not only IL-17-dependent EAE (*Komiyama et al., 2006*), but also several models of murine asthma in both Th1 and Th2-skewed backgrounds (*Gueders et al., 2009*). This initially appears at odds with the notion that administration of IL-4 in asthmatic mice aggravates disease (*Steinke and Borish, 2001*). However, we found that the combination of IL-2 and IL-4 develops Tregs with highly effective suppression of IL-4 production in Tconv cells in vitro (*Figure 3*). With the loss of IL-4-driven STAT6 phosphorylation in Tregs stimulated by both cytokines, we have demonstrated that the IL-4 signal leading to STAT6 activation is dramatically inhibited upon concurrent IL-2 exposure within Tregs. This data dovetails with many reports aligning IL-4 with the suppressive activity of Tregs (*Yang et al., 2017*; *Thornton et al., 2004*; *Pace et al., 2005*) and reinforces the notion that the functional outcome of multiple cytokines is highly context dependent and may not reflect the outcome of the cytokine in isolation, leading to pleiotropy in complex systems.

The dosing of the combinatorial cytokines in this study was based off of our previously published findings detailing CD4^+^FoxP3^-^CD45Rb^lo^ T effector/memory cell cytokine production in vitro following polysaccharide A (PSA) stimulation (*Jones et al., 2019*). Although IL-2 and IL-4 have independently shown efficacy in suppressing EAE (*Rouse et al., 2013*; *Racke et al., 1994*), the dosing used in this paper was substantially lower. For example, IL-2 limited EAE severity using an i.v. treatment regimen of 10,000 IU/dose, three times per day, for 3 days, totaling to 90,000 IU over the course of the trial (*Rouse et al., 2013*). Here, we use a total of 2500 IU of IL-2 over the course of the trial as a therapy. For IL-4, we administered 16.24 ng (1.2 pmol) of IL-4 per dose, every other day, whereas one commonly cited study utilized 1.0 µg (74.1 pmol) of IL-4 every 8 hr over the course of 11 days (*Racke et al., 1994*). At the low doses used in the present study, IL-2 and IL-4 alone were ineffective at suppressing disease (*Figure 7A and B*); however, when administered together, the cytokine combination significantly suppressed disease. This robust cytokine synergism translates into using far less cytokine to achieve therapeutic efficacy, potentially decreasing any unwanted side effects in clinical applications.

In the case of IL-2 and IL-4, the net functional result of the discovered pathway is a greatly enhanced population of immune inhibitory IL-10^+^ Tregs. The cellular activity was potent enough to

not only reduce disease prophylactically, but also reverse ongoing inflammation in both asthma and EAE models. It is therefore interesting to note that the use of both IL-2 and IL-4 appears to be a characteristic of some beneficial microbiota antigen responses. For example, *Bacteroides fragilis*, which is native to a healthy human gut (*Wexler and Goodman, 2017*; *Troy, 2010*), is known to regulate the peripheral immune system (*Johnson et al., 2018*; *Johnson et al., 2015a*; *Ochoa-Reparáz et al., 2010*; *Tzianabos et al., 1994*) and even benefit the gut-brain axis (*Sharon et al., 2019*; *Hsiao et al., 2013*) through T cell recognition of its capsular polysaccharide PSA (*Cobb et al., 2004*). It is now clear that PSA-specific effector T cells require communication with Tregs, and that immune suppression and inflammatory disease reversal depends upon both IL-4 (33, 45) and IL-2 (67), thereby providing a biological context in which the use of cytokines in combination has evolved to direct a healthy immune tone.

From a translational perspective, cytokine therapies have been limited in part due to the short half-life in circulation (*Donohue and Rosenberg, 1983*; *Conlon et al., 1989*). As such, our findings suggest that IL-2- and IL-4-induced combinatorial signaling may be best harnessed for autologous Treg transfer therapies, which itself has been limited by poor expansion and survival of Tregs ex vivo (*Tang and Bluestone, 2013*). Here, Tregs that underwent stimulation by both IL-2 and IL-4 together not only retained their activated and IL-10-producing state, but also exponentially expanded and survived for almost 2 weeks in culture with 3 days of cytokine stimulation and 9 days of co-culture with Tconv cells (*Figure 3M*). Moreover, the IL-2 and IL-4 combination appears to re-program Tregs with a receptor expression and STAT signaling pattern that facilitates and promotes continued suppressive capacity over extended periods of time through enhanced sensitivity to both cytokines.

In conclusion, we identified a novel mechanism of cell signaling integration controlling the transcriptional and proliferative response of regulatory T cells. While this is demonstrated in T cells using only two cytokines, these findings indicate that combinatorial signaling with other cytokines and cell types is possible, and is even beginning to emerge in the literature (*Karki et al., 2021*). Moreover, while these differences in response may underlie cytokine pleiotropy, they also provide an intriguing guide for new therapeutic applications of combinatorial cytokines in the clinical setting ranging from ex vivo support of autologous cell transfers to immune activation in cancer and inhibition for autoimmunity.

# Materials and methods

## Key resources table

| Reagent type (species) or resource | Designation | Source or reference | Identifiers | Additional information |
|---|---|---|---|---|
| Strain, strain background (*Mus musculus*) | C56Bl/6J | JAX | JAX 000664, RRID:IMSR_JAX:000664 | |
| Strain, strain background (*Mus musculus*) | *Il10*$^{-/-}$ | JAX | JAX 002251, RRID:IMSR_JAX:002251 | |
| Strain, strain background (*Mus musculus*) | *Il10*$^{fl/fl}$ | PMID:15534372, PMID:28783045 | from Dr. Werner Muller via Dr. Asma Nusrat | |
| Strain, strain background (*Mus musculus*) | *Foxp3*$^{creYFP}$ | JAX | JAX 016959, RRID:IMSR_JAX:016959 | |
| Strain, strain background (*Mus musculus*) | *Il10*$^{fl/fl}$ x *Foxp3*$^{creYFP}$ (IL-10 cKO) | This paper | JAX 016959, RRID:IMSR_JAX:016959 and from Dr. Werner Muller via Dr. Asma Nusrat | We crossed IL-10fl/fl mice obtained from Dr. Asma Nusrat with commercially available FoxP3-creYFP mice. To obtain, contact the Cobb lab. |
| Strain, strain background (*Mus musculus*) | BALB/c *Foxp3*$^{GFP}$ | JAX | JAX 006769, RRID:IMSR_JAX:006769 | |

*Continued on next page*

*Continued*

| Reagent type (species) or resource | Designation | Source or reference | Identifiers | Additional information |
|---|---|---|---|---|
| Strain, strain background (*Mus musculus*) | *Foxp3*^*RFP* | JAX | JAX 008374, RRID:IMSR_JAX:008374 | |
| Strain, strain background (*Mus musculus*) | *Il10*^*GFP* | JAX | JAX 008379, RRID:IMSR_JAX:008379 | |
| Strain, strain background (*Mus musculus*) | *Foxp3*^*RFP* x *Il10*^*GFP* (Xmas) | This paper | JAX 008374, RRID:IMSR_JAX:008374 and JAX 008379, RRID: IMSR_JAX:008379 | We crossed two commercially available mouse lines. To obtain, contact the Cobb lab. |
| Antibody | αCD4-magnetic beads (rat, monoclonal) | Miltenyi Biotec | 130-049-201, RRID:AB_2722753 | (1:50–1:100), (50 µL) |
| Antibody | αCD3 (Armenian hamster, monoclonal) | eBioscience | 16-0031-85, RRID: AB_468848 | (1:400), (50 µL) |
| Antibody | αCD28 (Syrian hamster, monoclonal) | eBioscience | 16-0281-82, RRID: AB_468921 | (1:400), (50 µL) |
| Antibody | CD4-PE (rat, monoclonal) | eBioscience | 12-0042-82, RRID: AB_465510 | (1:250–1:500), (50 µL) |
| Antibody | CD4-AF647 (rat, monoclonal) | Biolegend | 100530, RRID: AB_389325 | (1:250–1:500), (50 µL) |
| Antibody | CD4-BV421 (rat, monoclonal) | BD | 740007, RRID: AB_2739779 | (1:250–1:500), (50 µL) |
| Antibody | F4/80-BV711 (rat, monoclonal) | BD | 565612, RRID: AB_2734769 | (1:250–1:500), (50 µL) |
| Antibody | CD25-PECy7 (rat, monoclonal) | BD | 561780, RRID: AB_10893596 | (1:250–1:500), (50 µL) |
| Antibody | CD25-PE (rat, monoclonal) | eBioscience | 12-0251-82, RRID: AB_465607 | (1:250–1:500), (50 µL) |
| Antibody | CD124-BB700 (rat, monoclonal) | BD | 742172, RRID: AB_2871410 | (1:250–1:500), (50 µL) |
| Antibody | CD122-APC (rat, monoclonal) | BD | 564924, RRID: AB_2739009 | (1:250–1:500), (50 µL) |
| Antibody | CD132-BV421 (rat, monoclonal) | BD | 740039, RRID: AB_2739809 | (1:250–1:500), (50 µL) |
| Antibody | CD213a1-Biotin (rabbit, polyclonal) | MyBioSource | MBS2001753 | (1:250–1:500), (50 µL) |
| Antibody | pSTAT3-AF647 (mouse, monoclonal) | BD | 557815, RRID: AB_647144 | (1:100–1:200), (25 µL) |
| Antibody | pSTAT5-AF647 (mouse, monoclonal) | BD | 612599, RRID: AB_399882 | (1:100–1:200), (25 µL) |
| Antibody | pSTAT6-AF647 (mouse, monoclonal) | BD | 558242, RRID: AB_647145 | (1:100–1:200), (25 µL) |
| Antibody | IL-10R biotin (rat, monoclonal) | Biolegend | 112704, RRID: AB_313517 | (1:250–1:500), (50 µL) |
| Antibody | SA-APCefluor780 | eBioscience | 47-4317-82, RRID: AB_10366688 | (1:250–1:500), (50 µL) |
| Antibody | SA-BV605 | BD | 563260, RRID: AB_2869476 | (1:250–1:500), (50 µL) |
| Peptide, recombinant protein | Recombinant IL-2 | R and D Systems | 402 ML-020 | |
| Peptide, recombinant protein | Recombinant IL-4 | R and D Systems | 404 ML-010 | |
| Peptide, recombinant protein | Recombinant IL-10 | R and D Systems | 417 ML | |
| Commercial assay, kit | IFNγ ELISA MAX | Biolegend | 430803 | |
| Commercial assay, kit | IL-4 ELISA MAX | Biolegend | 431102 | |
| Commercial assay, kit | IL-10 ELISA MAX | Biolegend | 431413 | |
| Commercial assay, kit | IL-17A ELISA MAX | Biolegend | 432501 | |
| Commercial assay, kit | MOG35-55/CFA Emulsion PTX | Hooke Laboratories | EK-2110 | |
| Chemical compound, drug | STAT5i | Stemcell | 73852 | |

*Continued on next page*

*Continued*

| Reagent type (species) or resource | Designation | Source or reference | Identifiers | Additional information |
|---|---|---|---|---|
| Chemical compound, drug | House dust mite | Greer | XPB8103A2.5 | |
| Chemical compound, drug | Ovalbumin | Sigma-Aldrich | A5503 | |
| Chemical compound, drug | Cockroach antigen | Greer | XPB46D3A4 | |
| Chemical compound, drug | Methacholine | Sigma-Aldrich | 62-51-1 | |
| Software, algorithm | Prism | Graphpad | Versions 5 and 7, RRID: SCR_002798 | |
| Software, algorithm | FlowJo | FlowJo | v10.7, RRID:SCR_008520 | |
| Other | FACS-ARIA SORP | BD | | |
| Other | Attune NxT | Thermo Fisher | | |
| Other | Sytox Red | Invitrogen | S34859 | |
| Other | Fixable Viability Stain 510 | BD | 564406 | |
| Other | MOG I-A(b) tetramer-BV421 | NIH Tetramer Core | GWYRSPFSRVVH | |
| Other | Control I-A(b) tetramer-BV421 | NIH Tetramer Core | PVSKMRMATPLLMQA | |
| Other | CellTrace Violet | Invitrogen | C34557 or C34571 | |
| Other | Europium-conjugated streptavidin | Perkin-Elmer | 1244–360 | |
| Other | DELFIA Enhancement solution | Perkin-Elmer | 4001–0010 | |
| Other | 10% formalin | Anachemia | 41916–700 | |
| Other | Hematoxylin | Ricca | 3536–32 | |
| Other | Eosin | EMD Harleco | 588X-75 | |
| Other | Solvent Blue 38 | Sigma-Aldrich | S3382 | |
| Other | Percoll | GE | 17-0891-02 | |

## Mice

C57BL/6J (Stock #000664), $Il10^{GFP}$ (B6.129S6-$Il10^{tm1Flv}$/RthsnJ, Stock #008379), $Foxp3^{RFP}$ (C57BL/6-$Foxp3^{tm1Flv}$/J, Stock #008374), $Foxp3^{creYFP}$ (B6.129(Cg)-$Foxp3^{tm4(YFP/icre)Ayr}$/J, Stock 016959), and $Il10^{-/-}$ (B6.129P2-$Il10^{tm1Cgn}$/J, Stock #002251) mice, all on the C57BL/6 background, as well as $Foxp3^{GFP}$ mice on the BALB/c background (C.Cg-$Foxp3^{tm2Tch}$, Stock #006769) were purchased from the Jackson Laboratory (Bar Harbor, ME). $Il10^{fl/fl}$ mice were kindly gifted by Dr. Werner Muller (*Roers et al., 2004*) via Dr. Asma Nusrat (*Quiros et al., 2017*). $Il10^{GFP}$ and $Foxp3^{RFP}$ mice were crossed to make $Il10^{GFP}$ x $Foxp3^{RFP}$ dual-reporters, and $Il10^{fl/fl}$ and $Foxp3^{cre-YFP}$ were crossed to make FoxP3-specific IL-10 knockouts (IL-10 cKO) in our facility. Mice were housed in a 12 hr light/dark cycle-specific pathogen-free facility and fed standard chow (Purina 5010) ad libitum. Enrichment and privacy were provided in mating cages by 'breeding huts' (Bio-Serv S3352-400). Mouse studies and all animal housing at Case Western Reserve University were approved by and performed according to the guidelines established by the Institutional Animal Care and Use Committee of CWRU.

## Primary splenic T cells

Primary splenocytes were isolated from freshly harvested mouse spleens, and reduced to a single cell suspension by passing them through a sterile 100 µM nylon mesh cell strainer (Fisher Scientific, Hampton, NH). The single cell suspensions were labeled with anti-mouse CD4 magnetic microbeads (Miltenyi Biotec, San Diego, CA) and separated with an AutoMACS Pro Separator (Miltenyi Biotec) per manufacturer's instructions.

## Cell culture

After flow sorting, Tregs or Tconv cells were cultured in 96-well round bottom plates (Corning, Corning, NY) at 50,000 cells per well in advanced RPMI (Gibco/Fisher Scientific, Waltham, MA)

supplemented with 5% Australian-produced heat-inactivated fetal bovine serum, 55 µM β-mercaptoethanol, 100 U/mL and 100 µg/mL Penicillin/Streptomycin, and 0.2 mM L-glutamine (Gibco/Fisher Scientific, Waltham, MA) at 5% $CO_2$, 37°C. For activating conditions, wells were coated with αCD3ε (eBioscience, San Diego, CA) at 2.5 µg/mL in PBS and then incubated at 4°C overnight followed by two washes with PBS before receiving cells. As indicated, cultures were supplemented with αCD28 (eBioscience), recombinant IL-2 (R and D Systems, Minneapolis, MN), IL-4 (R and D Systems), and IL-13 (Invitrogen, Carlsbad, CA), with equimolar concentration being 728 pM. For conditions receiving two or more cytokines, the total concentration of cytokines in the culture well is equal in fold to the number of cytokines indicated (e.g. IL-2 = onefold cytokine, IL-2/IL-4 = twofold cytokine). IL-10R-biotin (Biolegend, San Diego, CA) and recombinant IL-10 (R and D Systems) were also supplemented as indicated. For experiments involving STAT5 inhibitor, STAT5i (Stemcell, Vancouver, BC, CAN) resuspended in DMSO was added directly to culture at the designated amounts. Due to the known toxicity of DMSO, controls with DMSO only were also included, with percent inhibition calculated compared to the DMSO controls.

## In vitro *Tconv suppression*

Tregs were pre-stimulated with αCD3ε and all combinations of IL-2 and IL-4 for 3 or 7 days, washed, then mono-cultured or introduced into co-culture with 50,000 freshly isolated and CellTrace-stained wild-type Tconv cells at a 1:2, 1:4, 1:8, 1:16, 1:32, and 1:64 dilution (*Collison and Vignali, 2011*). To assess the suppressive ability of a Treg on a per cell basis, Tregs were counted after stimulation and before placing in co-culture. To determine the suppressive ability of Tregs on a population level, Tregs were counted prior to stimulation so that the number of cells introduced into co-culture with Tconv reflected the proliferation induced by the cytokine conditions. CellTrace signal in Tconv cells was detected by flow cytometry after 3 days of co-culture, and secretion of cytokines was quantified by ELISA.

## EAE model of multiple sclerosis

Age-matched female mice between 11 and 23 weeks underwent EAE induction using the $MOG_{35-55}$/CFA Emulsion and pertussis toxin (PTX) kit from Hooke Laboratories (Lawrence, MA) according to manufacturer's instructions. Mice were immunized with 200 µg of $MOG_{35-55}$/CFA emulsion s.c. in two locations in the back on day 0 and were administered 100 ng of PTX i.p. on days 0 and 1 of the trial. Negative controls for disease received a CFA emulsion and PTX according to the same schedule. Mice were weighed and scored daily according to Hooke Laboratory's EAE scoring guide (https://hookelabs.com/protocols/eaeAI_C57BL6.html). Treated mice received cytokine cocktail comprising of equimolar IL-2 (20 ng) and IL-4 (16.24 ng) i.v. in a volume of 100 µL of sterile PBS every other day. Untreated control mice received 100 µL of sterile PBS i.v. at the same time intervals.

## HDM and TAC models of pulmonary inflammation

Age- and sex-matched mice between 8 and 23 weeks were challenged with HDM antigen (HDM, *D. Farinae*, GREER, Lenoir, NC) by i.n. delivery of 20 µg HDM/dose in PBS according to the designated challenge schedule (*Figures 5A,C* and *6A*) and sacrificed accordingly (*Johnson et al., 2004*). IL-2 and IL-4 at the designated doses and time intervals were delivered i.n. in a mixed cocktail according after animals were anesthetized with 3% isoflurane (Baxter, Deerfield, IL) with an anesthesia system (VetEquip, Livermore, CA). In the chronic triple antigen combination model, mice were sensitized intraperitoneally with 20 µg Ovalbumin (OVA, Albumin from chicken egg white, Millipore Sigma, Darmstadt, Germany), 2.5 µg cockroach antigen (CRA, American, *Periplaneta Americana*, GREER, Lenoir, NC), and 2.5 µg of HDM in PBS. Immediately following sensitization, mice were challenged intranasally with HDM, CRA, or OVA in a rotating schedule for 7 weeks (*Duechs et al., 2014*). For Flexivent (SCIREQ, Montreal, Quebec) studies, mice were anesthetized with a cocktail of Ketamine/Xylazine/Acepromazine, the trachea was exposed and catheterized, and methacholine was administered via nebulization in increasing doses. Airway resistance was measured on a Flexivent respirator over time. After mice were euthanized, bronchial alveolar lavage fluid (BALf) was recovered in three lavages of 1 mL each, and lung tissues underwent FFPE processing or were reduced to a single cell

suspension for flow cytometric analysis. BALf cell differentials were acquired by a HemaVet 950 Hematology Analyzer.

## Flow cytometry and cell sorting

For splenic Treg cell sorting, magnetic bead-mediated positively selected CD4$^+$ cells were sorted by endogenously expressed FoxP3$^{RFP}$ and/or IL-10$^{GFP}$. For IL-10$^{-/-}$ mice, CD4$^+$ cells were stained and sorted with CD25-PE-Cy7 (BD Biosciences, San Jose, CA). Cells were washed in MACS buffer (Miltenyi Biotec) before sterile cell sorting using a FACSAria-SORP (BD Biosciences). For flow cytometric analysis of tissue compartments, cells were passed through a sterile 70 µM nylon mesh cell strainer (Fisher Scientific, Hampton, NH) for homogenization into a single cell suspension. Blood and spleen were RBC-lysed (BD, Franklin Lakes, NJ), and brain and spinal underwent a 30%/70% Percoll gradient centrifugation (GE, Boston, MA). For intracellular staining, cells were fixed and permeabilized with the FoxP3/transcription factor staining buffer set (eBioscience). Commercial antibodies used included: CD4-PE (eBioscience), CD4 – AF647 (Biolegend), F4/80 – BV711 (BD), CD25-PE (eBioscience), CD25 – PE-Cy7 (BD), CD124 – BB700 (BD), CD122 – APC (BD), CD132 – BV421 (BD), CD213a1 – Biotin (MyBioSource, San Diego, CA), IL-10R-biotin (Biolegend), streptavidin – BV605 (BD), streptavidin – APCefluor780 (eBioscience), pSTAT3 – AF647 (BD), pSTAT5 – AF647 (BD), and pSTAT6 – AF647 (BD). Additional flow cytometry reagents purchased commercially include: Sytox Red (Invitrogen), CellTrace Violet (Invitrogen), and Fixable Viability Stain 510 (BD). Tetramers obtained from the NIH Tetramer Core include MOG$_{35-55}$ Tetramer – BV421, I-A(b) Tetramer – BV421 (Atlanta, GA). Cells were acquired on an Attune NxT cytometer (Invitrogen). Cytometer usage was supported by the Cytometry and Imaging Microscopy Core Facility of the Case Comprehensive Cancer Center. Analysis of all FACS data was performed using FlowJo v10 (Tree Star, Inc, Ashland, OR). Analysis of proliferation was done in FlowJo v10 using the Proliferation Modeling platform.

## Cell proliferation

Cell proliferation was quantified using CellTrace Violet staining coupled with analysis of dilution using flow cytometry. Division Indices were calculated mathematically using the Proliferation Modeling platform in Flowjo by dividing the total number of cell divisions by the total number of cells in the starting culture. Proliferation indices were calculated by dividing the total number of divisions by the number of cells that underwent division.

## ELISA and Luminex

Cytokine concentrations were analyzed by sandwich ELISA performed according to manufacturer's instructions (BioLegend). The assay was modified to utilize europium-conjugated streptavidin (Perkin-Elmer, San Jose, CA). Signal was detected with a Victor V3 plate reader (Perkin Elmer). For Luminex assays, media were snap frozen in liquid nitrogen and sent to Eve Technologies (Calgary, AB) for mouse 31-plex and TGFβ 3-plex analysis.

## Histology and microscopy

Harvested tissues were fixed in 10% formalin (VWR, Radnor, PA) for 24 hr and sent to AML Laboratories (Jacksonville, FL) for paraffin embedding. Embedded lung tissues were sectioned and stained with H and E, Periodic Acid Schiff (PAS), or trichrome by AML and spinal cord tissues were sectioned and stained with Hematoxylin and Eosin (H and E) (Ricca Chemical, Arlington, TX and Sigma Aldrich, St. Louis, MO) or Luxol Fast Blue (LFB) (Sigma Aldrich) in our lab. Images were acquired with a Leica DM IL LED (Wetzlar, Germany).

## Power analyses

Hooke Laboratories established that 10–12 mice are needed in MOG challenge groups to account for the natural scatter of this preclinical disease model, which is supported by our own power calculations. For these calculations, effect size was conservatively set to 1.25, and the α significance (within the 95% confidence interval) at 0.05 using the online calculator at http://euclid.psych.yorku.ca/SCS/Online/power/. Using these analyses, we also established that six mice were necessary for our asthma trials, and biological triplicates are necessary for in vitro Treg stimulation assays.

## Data analyses

All data are represented by mean ± SEM. Pulmonary inflammation data sets include a minimum of four, but more commonly six animals per group per experiment. EAE trials include four animals in the negative disease control conditions and 10–12 animals in groups that receive MOG peptide. Data and statistical measurements were generated with Prism (Graphpad, San Diego, CA). For comparisons between two groups, Student's t-test was used; comparisons between multiple groups utilized analysis of variance. The PCA was generated in RStudio using factoextra and ggplot2. Mean ± SEM are indicated. *p<0.05, **p<0.01, ***p<0.001, ****p<0.0001.

## Acknowledgements

We are immensely grateful for the help of Douglas M Oswald, PhD, for valuable scientific input and guidance in R, Mark B Jones, PhD, for mentorship and project discussions early on, Kalob M Reynero for technical assistance in tissue sectioning, Jill M Cavanaugh for technical assistance and maintenance of the mouse colony, and Lori. SC Kreisman for laboratory support and manuscript editing. Moreover, we thank Sandra Siedlak and Xiongwei Zhu, PhD, for expertise in histological sectioning and staining, and Alex Huang, MD, PhD, for equipment for histological imaging. We thank the Cytometry and Microscopy Shared Resource of the Case Comprehensive Cancer Center (P30CA043703) for equipment and assistance with flow cytometry-based experiments. This work was made possible by grants from: The National Institutes of Health (R01-GM115234 and R01-AI154899), and the Hartwell Foundation to BAC, and the National Institutes of Health (T32-AI089474) to JYZ and CAA.

## Additional information

### Funding

| Funder | Grant reference number | Author |
| --- | --- | --- |
| National Institutes of Health | GM115234 | Brian A Cobb |
| National Institutes of Health | AI154899 | Brian A Cobb |
| National Institutes of Health | AI089474 | Julie Y Zhou<br>Carlos A Alvarez |
| Hartwell Foundation | | Brian A Cobb |

The funders had no role in study design, data collection and interpretation, or the decision to submit the work for publication.

### Author contributions

Julie Y Zhou, Conceptualization, Formal analysis, Investigation, Methodology, Writing - original draft, Writing - review and editing; Carlos A Alvarez, Investigation; Brian A Cobb, Conceptualization, Resources, Formal analysis, Supervision, Funding acquisition, Methodology, Writing - review and editing

### Author ORCIDs

Julie Y Zhou ⬮ https://orcid.org/0000-0002-1533-5183
Brian A Cobb ⬮ https://orcid.org/0000-0003-1055-2530

### Ethics

Animal experimentation: This study was performed in strict accordance with the recommendations in the Guide for the Care and Use of Laboratory Animals of the National Institutes of Health. All of the animals were handled according to approved institutional animal care and use committee (IACUC) protocol (#2014-0047) of Case Western Reserve University School of Medicine.

Decision letter and Author response

Decision letter https://doi.org/10.7554/eLife.57417.sa1

Author response https://doi.org/10.7554/eLife.57417.sa2

## Additional files

### Supplementary files

• Transparent reporting form

### Data availability

All data generated or analysed during this study are included in the manuscript and supporting files.

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
