## [Decision Letter]

**Acceptance summary:**

While it is well established that Tregs rely on IL-2 for their function, how combination of other cytokines contribute to their differentiation and function is less well understood. Zhou and colleagues demonstrate that IL-4 may, in combination with IL-2, support the proliferation of Tregs and their IL-10 production, and suggest that IL-4 may be used to help treat autoimmunity and allergy by either direct administration with IL-2 or by ex-vivo expansion of Tregs. This report thus shows a novel effect of combined IL-2 and IL-4 signaling on IL-10^+^ Tregs.

**Decision letter after peer review:**

Thank you for submitting your article "Integration of IL-2 and IL-4 Signals Coordinates Divergent Regulatory T cell Responses and Drives Therapeutic Efficacy" for consideration by *eLife*. Your article has been reviewed by three peer reviewers, one of whom is a member of our Board of Reviewing Editors, and the evaluation has been overseen by Tadatsugu Taniguchi as the Senior Editor. The following individual involved in review of your submission has agreed to reveal their identity: Onur Boyman (Reviewer #2).

The reviewers have discussed the reviews with one another and the Reviewing Editor has drafted this decision to help you prepare a revised submission.

Summary:

Given the role of Tregs in autoimmunity and tumor responses, increasing interest has focused on how these cells can be manipulated to improve their suppressive function. While it is well established that Tregs rely on IL-2 for their function, how combination of other cytokines contribute to their differentiation and function is less well understood. Zhou and colleagues demonstrate that IL-4 may, in combination with IL-2, support the proliferation of Tregs and their IL-10 production, and suggest that IL-4 may be used to help treat autoimmunity and allergy by either direct administration with IL-2 or by ex-vivo expansion of Tregs. This report thus shows a novel effect of combined IL-2 and IL-4 signaling on IL-10^+^ Tregs.

Essential revisions:

1) The in vitro effects largely focused on the effect on Treg cells, while the in vivo experiment failed to show the Treg-specific in vivo relevance. The IL-10^-/-^ mice that were used as control have been described with chronic colitis conditions that may bring additional caveats due to the difference in cellular activation at steady state. Therefore, more functional and mechanistic studies are required to show that the combinatorial effect of IL-2/IL-4 is acting through Treg cells. In Figure 1, more IL-10 production comes from 36-72hr of treatment. is the co-treatment of IL-2 and IL-4 only essential in later time of culturing, and initial activation is sufficient with IL-2 or IL-4 alone?

The authors reported higher proliferative capacity and viability from IL-10^+^ Treg cells with dual IL-2/IL-4 treatment. Does IL-10 cytokine itself stimulate Treg cell proliferation and IL-10 production autonomously? Would the effect be abrogated by anti-IL-10?

2) In Figure 2, the levels of Treg proliferation are very low. The authors used plate-bound anti-CD3 alone together with or without a relatively low concentration of IL-2. The culture conditions seem to be the same throughout the manuscript. Have the authors performed dosing experiments to see if the Tregs are given more IL-2, or anti-CD28 etc., the same effects are seen? There is a concern that the culture conditions may be suboptimal so that much of the in vitro data can be interpreted as IL-4 rescuing Tregs in these low proliferation, low survival conditions. There may be a case for using limiting conditions in some settings to model in vivo conditions with limited stimuli but the interpretation would be somewhat different.

In addition it is unclear whether the concentration of IL-2 + IL-4 was 2x, i.e. double the concentration of the single cytokine conditions. Please complete this information in the figure legend and the Materials and methods.

3) In Figure 3, the authors claim that IL-2/IL-4 increase Treg suppressive activity. However, it is difficult to see this in Figure 3A-F, as the level of suppression appears to be the same between the IL-2 and IL-2/IL4 groups. In Figure 3G-J some differences do emerge but this may likely reflect changes in a poor condition of the Tregs after prolonged culture with IL-2/IL-4. The authors should consider if what they are seeing here is a real gain of suppressive function or a retention of suppressive function/Treg survival after prolonged culture.

The suppressive assay in Figure 3 showed that there is no increased suppressive capacity when Treg cell number was normalized prior to assay, but when the increased proliferative capacity were accounted for without pre-normalization of cell numbers, there was an increase in suppression. This was, however, contradicting with the hypothesis of IL-10 dependency. IL-2/IL-4 treatment should have increased %IL-10 producing Treg cells, and would have resulted in increased suppression in Figure 3A if IL-10 is the key dependent factor.

Also, IL-10 KO mice have been reported with chronic colitis, and may bring additional caveats to the in vitro suppressive assay conducted in Figure 3. Are the conventional T cells from the IL-10^-/-^ mice? Do the Treg cells from IL-10^-/-^ mice have altered functionality to begin with? Would treatment with anti-IL-10 give similar results?

To pinpoint that there is specific IL-2/IL-4 dual effect specific for Treg cells, the authors should consider Treg cell specific KO of IL-4R and see if observed effects are abolished.

4) In Figure 4, STAT5 inhibitor affects general Treg cell viability, is this IL-10 specific or inhibitor specific? If treated with another irrelevant inhibitor (ex: STAT6i), would it have similar effect of decreasing IL-10?

In Figure 4C, why does IL-2 alone not induce pSTAT5? What is the concentration of IL-2 used here? There is a clear synergistic effect of IL-2 and IL-4 but again the lack of a baseline response to IL-2 is a concern. Dose response experiments would help immensely. Please also show isotype staining to insure minimal background staining effect and fair comparison.

Figure 4A-C, it is unclear whether the heatmaps represent percentages or MFIs. Please complete this information in the figure legend.

5) In Figure 5, does IL-10R expression change play a role in the synergistic increase of IL-10?

6) In Figure 6A, what were the cell types analyzed? The legend did not describe them.

In the house dust mite experiment, is the protective effect Treg cell-specific? For instance, alveolar macrophages and Th2 cells are known to take roles in the house dust mite-driven asthma model, and may also be producers of IL-10 in certain context. Does IL-4 and IL-2 treatment alter these cellular population, phenotype and infiltration that may attribute to the observed histology?

In Figure 6D, many of the cell number figures lack a comparison between IL-4 alone at a high dose. Is IL-2 needed? The authors show data of IL-10 and Tregs from separate groups but all the total cell number analysis is combined.

7) In Figure 7, the EAE experiments lack comparison between IL-2/IL-4 and either of these cytokines alone. IL-2 in particular has been used in many EAE experiments with varied effects, without some indication that the IL-2/IL4 combination has differential effects to the separate cytokines. Any of this data is hardly interpretable.

Previous article suggested that systemic administration of IL-4 and IL-2 alone improves EAE, possibly through DCs and Treg cells, respectively. Is there data to suggest the in vivo phenotype observed in Figure 7 is a synergy with IL-4 and IL-2? Or are they offering similar degree of protection against EAE disease development?

Do conventional T cells express IL-10 in asthma model and EAE model in vivo? Would they attribute to the loss of effect in IL-10^-/-^ model?

8) Several other reports have looked at the role of IL-4 in Treg function and demonstrated its role in supporting Treg proliferation and suppressive function in both mice and humans. The current Discussion should include a greater overview of the current literature.

Thornton, Piccirillo and Shevach, 2004; Pace, Pioli and Doria, 2005; Yang et al., 2017.

---

## [Author Response]

Essential revisions:1) The in vitro effects largely focused on the effect on Treg cells, while the in vivo experiment failed to show the Treg-specific in vivo relevance. The IL-10^-/-^ mice that were used as control have been described with chronic colitis conditions that may bring additional caveats due to the difference in cellular activation at steady state. Therefore, more functional and mechanistic studies are required to show that the combinatorial effect of IL-2/IL-4 is acting through Treg cells.

We have added data from new experiments involving a FoxP3-specific IL-10 knockout mouse (IL10^fl/fl^-FoxP3^creYFP^, termed IL-10 cKO in the manuscript), including validation that the Tregs do not make IL-10 (Figure 3H, Figure 3—source data 1C), a Treg suppression assay (Figure 3G-I, Figure 3—source data 1C), asthma trial (Figure 6D-F), and EAE trial (Figure 7I, Figure 7—figure supplement 7B). All new experiments indicate that IL-10 production by Tregs is necessary for in vitro and in vivo suppression of disease induced by IL-2/IL-4.

Pertaining to the spontaneous development of colitis in IL-10^-/-^ mice, we have added colon lengths, which is a common measure of colitis, of IL-10^-/-^, IL-10 cKO, and WT mice in our colony. These data suggest that the mice at the age of sacrifice do not have active colitis (Figure 3—source data 1C–D). According to JAX website for the IL-10^-/-^ mouse used in our study (Stock No: 00225):

“The onset and severity of both spontaneous and experimentally-induced inflammatory phenotype of IL10-deficient mice is strongly influenced by the genetic background and the husbandry conditions (specific health status/commensal flora) of the vivarium in which mice are maintained.”

We have not observed colitis in our colony for the years that we have maintained the IL-10^-/-^ mouse. We have edited the language in our manuscript to reflect the confounding factors that colitis may introduce. We also hope that by including a second and cell type-specific model of IL-10 ablation, our results will be more convincing, especially since no intestinal pathology was seen is these mice.

In Figure 1, more IL-10 production comes from 36-72hr of treatment. is the co-treatment of IL-2 and IL-4 only essential in later time of culturing, and initial activation is sufficient with IL-2 or IL-4 alone?

We expanded the sequential cytokine experiment in Figure 1 to include incubation of Tregs with all cytokine conditions for 36 hours, then a switch to all other cytokine conditions for the remainder of the 72 hr. (3 day) incubation (Figure 1H). In the early culture period (0-36 hr.), IL-2/IL-4 supplemented Tregs have already produced more IL-10 than the other cytokine supplementation conditions. During the later culture period (37-72 hr.), IL-2/IL-4-supplementation also leads to greater IL-10 production by Tregs. The presence of IL-2/IL-4 appears to be necessary during all phases of the 72-hour time course.

The authors reported higher proliferative capacity and viability from IL-10^+^ Treg cells with dual IL-2/IL-4 treatment. Does IL-10 cytokine itself stimulate Treg cell proliferation and IL-10 production autonomously? Would the effect be abrogated by anti-IL-10?

Using our newly created FoxP3-specific IL-10 knockout (IL-10cKO), we found that in the absence of IL-10, there is no longer IL-2/IL-4-enhanced Treg proliferation (Figure 2G). Combined with our observation that IL-10^+^ Tregs have higher proliferative capacity with IL-2/IL-4 supplement (Figure 2D, F), our data supports a model in which IL-10 production is linked to enhanced proliferation of Tregs upon stimulation with IL-2 and IL-4 in combination.

2) In Figure 2, the levels of Treg proliferation are very low. The authors used plate-bound anti-CD3 alone together with or without a relatively low concentration of IL-2. The culture conditions seem to be the same throughout the manuscript. Have the authors performed dosing experiments to see if the Tregs are given more IL-2, or anti-CD28 etc, the same effects are seen? There is a concern that the culture conditions may be suboptimal so that much of the in vitro data can be interpreted as IL-4 rescuing Tregs in these low proliferation, low survival conditions. There may be a case for using limiting conditions in some settings to model in vivo conditions with limited stimuli but the interpretation would be somewhat different.

This question prompted us to re-evaluate our flow data. In so doing, we altered the gating strategy to better identify live Tregs within each dataset. In the original version, we included too many signals that were CellTrace negative; therefore, in the revised manuscript, we excluded a population of signals that were too low on the FSC/SSC plot to be considered live Tregs. In so doing, the number of dividing Tregs, as a percentage, increased substantially. The patterns we reported originally did not change. Likewise, the interpretation does not change. Nonetheless, we believe that the data more accurately represents what’s actually happening in each experiment. The revised gating is provided (Figure 2—figure supplement 1A) and is used throughout all relevant data and figures in this revised submission.

To directly address the reviewer’s concern that the IL-2 concentrations in our experiments are suboptimal, we stimulated Tregs with an IL-2 concentration gradient ranging from 0.01x to 100x of the dose used (Figure 1—figure supplement 2A, Figure 2—figure supplement 1B). Increasing the concentration of IL-2 markedly does not alter the observed phenomenon of synergistic IL-10 production and enhanced Treg proliferation due to the IL-2/IL-4 combination.

We also evaluated whether the addition of co-stimulation (plate-bound αCD28) to Treg cultures changes the phenotype resulting from IL-2/IL-4 stimulation. It did not (Figure 1—figure supplement 2B, Figure 2—figure supplement 1A).

In addition it is unclear whether the concentration of IL-2 + IL-4 was 2x, i.e. double the concentration of the single cytokine conditions. Please complete this information in the figure legend and the Materials and methods.

The concentration of IL-2/IL-4 is double the concentration of the single cytokine conditions. The relevant figure legends and the Materials and methods have been edited to include this information.

3) In Figure 3, the authors claim that IL-2/IL-4 increase Treg suppressive activity. However, it is difficult to see this in Figure 3A-F, as the level of suppression appears to be the same between the IL-2 and IL-2/IL4 groups. In Figure 3G-J some differences do emerge but this may likely reflect changes in a poor condition of the Tregs after prolonged culture with IL-2/IL-4. The authors should consider if what they are seeing here is a real gain of suppressive function or a retention of suppressive function/Treg survival after prolonged culture.

Poor condition of the Tregs after prolonged culture is a possible interpretation, but we do not believe this is the case. The frequency of IL-10^+^ Tregs increases over the course of 3 to 7 days, with IL-10^+^ Tregs accounting for >90% of the culture at day 7 (Figure 1L). Furthermore, we see an increase in the number of FoxP3^+^ cells in culture, as well as an increase in cell viability, IL-10 MFI, and IL-10 production over time (Figure 1—figure supplement 4B, Figure 2—figure supplement 1B). Combined with the observation that IL-10 expression increases with Treg division, which is enhanced by IL-2/IL- 4 (Figure 2C), our interpretation is that the Tregs are thriving, producing a lot of IL-10, are robustly suppressing Tconv cells, and the increase in suppressive behavior of the population is due to the combination of IL-10^+^ Treg proliferation and IL-10 release.

The suppressive assay in Figure 3 showed that there is no increased suppressive capacity when Treg cell number was normalized prior to assay, but when the increased proliferative capacity were accounted for without pre-normalization of cell numbers, there was an increase in suppression. This was, however, contradicting with the hypothesis of IL-10 dependency. IL-2/IL-4 treatment should have increased %IL-10 producing Treg cells, and would have resulted in increased suppression in Figure 3A if IL-10 is the key dependent factor.

In the WT suppression assays (Figure 3A-C), we found that inhibition of cytokine production was enhanced by the IL-2/IL-4 combination treatment of the Tregs prior to co-culture with Tconv cells compared to either cytokine alone, but that changes in proliferation were not different between IL-2 and IL-2+IL-4 (but were different from IL-4 alone). Using both complete IL-10^-/-^ and FoxP3-specific IL-10 cKO mice/Tregs (Figure 3D-I), however, we found a complete loss of all inhibitory activity (cytokine production and proliferation).

While the reviewer is correct in pointing out that the proliferation inhibition in Figure 3A does not perfectly track with IL-10 production in Figure 3B, we interpret this as meaning that inhibition of cytokine production (IL-4 especially – see Figure 3C) is more sensitive to IL-10-mediated inhibition. Our reasoning is that all IL-10 is washed away after the initial 3-day stimulation with IL-2, IL-4 or both in order to setup the co-cultures with Tconv cells. This means that the amount of IL-10 in the initial co-cultures is zero. It is true that the Tregs continue to produce IL-10, but the amount reported in Figure 3B is the accumulated IL-10 over the co-culture period. We believe that the lag time in reaching higher IL-10 concentrations disconnects the final IL-10 value from the proliferation measured at the end of the assay, whereas the cytokine production is more easily inhibited – hence the more robust effect. This is speculation, but considering the complete absence of inhibitory activity in both IL-10-deficient systems, we feel confident in our conclusion that the protective efficacy of IL-2/IL-4 combination therapy in our mouse models of disease is dependent upon IL-10.

Also, IL-10 KO mice have been reported with chronic colitis, and may bring additional caveats to the in vitro suppressive assay conducted in Figure 3. Are the conventional T cells from the IL-10^-/-^ mice? Do the Treg cells from IL-10^-/-^ mice have altered functionality to begin with? Would treatment with anti-IL-10 give similar results?

The conventional T cells used in all of the suppression assays are from wild type FoxP3-RFP mice to enable Treg exclusion via flow sorting FoxP3- T cells. The Tregs are the only cells in the assay that originate from the indicated strains. We have edited the manuscript to make this clearer. As mentioned in our response to question 1, we have added colon lengths of IL-10^-/-^, IL-10 cKO, and WT mice in our colony, suggesting that the mice at the age of sacrifice do not have active colitis (Figure 3—source data 1C–D). We have edited the language in our manuscript to reflect the confounding factors that colitis may introduce. Moreover, we have added the FoxP3-specific IL-10 knockout mouse/cells (which also do not develop colitis in our vivarium) to our dataset, which we believe further supports our conclusions.

To pinpoint that there is specific IL-2/IL-4 dual effect specific for Treg cells, the authors should consider Treg cell specific KO of IL-4R and see if observed effects are abolished.

Adding yet another novel mouse strain and the substantial number of relevant experiments in this manuscript is simply beyond our resources and not necessary to reach the conclusions reported herein. Our previous paper (https://doi.org/10.1371/journal.pone.0216893) identified that IL-4 signaling is central to polysaccharide A-expanded T effector memory cells, which are suppressive, but only through cooperating with Tregs. PSA-expanded Tem cells secrete IL-4, which is the necessary communicating factor to drive Tregs to produce IL-10. That paper includes an IL-4 KO model, in which the ability of Tems to robustly boost the IL-10 production of Tregs was abolished.

4) In Figure 4, STAT5 inhibitor affects general Treg cell viability, is this IL-10 specific or inhibitor specific? If treated with another irrelevant inhibitor (ex: STAT6i), would it have similar effect of decreasing IL-10?

We tested both STAT6i (AS1517499, Sigma) and STAT3i (VII, Sigma). We chose these specific inhibitors and their respective doses based off a combination of their utilization in existing literature

(doi: 10.1038/mi.2017.21, doi: 10.1189/jlb.0313154, doi: 10.1189/jlb.5A1114-532RR, doi: 10.1074/jbc.M113.475277) as well as their known specificity in not targeting STAT5. In our hands, both inhibitors appeared extremely toxic to Tregs as measured by Sytox exclusion in flow cytometry, much more than STAT5i (see Author response image 1). We were therefore not able to run the analyses as planned.

In Figure 4C, why does IL-2 alone not induce pSTAT5? What is the concentration of IL-2 used here? There is a clear synergistic effect of IL-2 and IL-4 but again the lack of a baseline response to IL-2 is a concern. Dose response experiments would help immensely. Please also show isotype staining to insure minimal background staining effect and fair comparison.

We added a dose response experiment showing that higher IL-2 concentrations markedly increased pSTAT5 expression (Figure 4—figure supplement 1A). This data demonstrates the potent cytokine synergy by enabling robust STAT5 phosphorylation at low IL-2 concentrations which do very little in typical cell culture experiments. We also added isotype staining to the existing figure panels (Figure 4A-C).

Figure 4A-C, it is unclear whether the heatmaps represent percentages or MFIs. Please complete this information in the figure legend.

The heatmaps represent MFIs. This information has been added to the figure legend.

5) In Figure 5, does IL-10R expression change play a role in the synergistic increase of IL-10?

In new experiments, αCD3 activation increased the expression of IL-10R, with IL-2 stimulation leading to the greatest increase (Figure 5I). IL-10R blockade decreased the expression of IL-10 in Tregs and obliterated the IL-2/IL-4-induced synergy measured by IL-10 production (Figure 5J), suggesting that there is some degree of autocrine signaling occurring through the IL-10/IL-10R axis in Tregs.

6) In Figure 6A, what were the cell types analyzed? The legend did not describe them.

In Figure 6A, we analyzed lymphocytes, monocytes, and granulocytes using a HemaTrue analyzer. We have clarified the cell types in the legend.

In the house dust mite experiment, is the protective effect Treg cell-specific? For instance, alveolar macrophages and Th2 cells are known to take roles in the house dust mite-driven asthma model, and may also be producers of IL-10 in certain context. Does IL-4 and IL-2 treatment alter these cellular population, phenotype and infiltration that may attribute to the observed histology?

We further investigated the Treg-specificity of IL-2/IL-4 therapy in the house dust mite model using the IL-10^fl/fl^ x FoxP3^creYFP^ (IL-10 cKO) mouse. In these mice, IL-2/IL-4 does not have a therapeutic effect seen through cellular infiltration into the bronchiolar space or histology (Figure 6D-F). It is possible that other cells provide IL-10, but when Tregs lack the ability to make IL-10, IL-2/IL-4 is not able to reach therapeutic efficacy. Furthermore, we included analyses of the IL-10 expression of CD4- cells in the lungs and spleens (Figure 6—figure supplement 1B). It does not appear that IL-2/IL4 increases the IL-10 expression in these cells as a population.

In Figure 6D, many of the cell number figures lack a comparison between IL-4 alone at a high dose. Is IL-2 needed? The authors show data of IL-10 and Tregs from separate groups but all the total cell number analysis is combined.

The original Figure 6D is now 6C. The data shown in Figure 6A-D is a *therapeutic* dosing of IL-2, IL-4, and IL-2/4 combination, meaning therapy was started after disease induction. In that setting, which uses a 1:1 ratio of IL-2 and IL-4 (1x each), we show the individual cytokine controls (see Figure 6A-C). In the revised Figure 6D, we abbreviate the presentation to just the combination to make two points. First, the absence of IL-10 leads to a loss of therapeutic efficacy. Second, even in a Th2-skewed asthma system (BALB/c), the combination of IL-2 and IL-4 reduces disease severity.

The “high dose” (10x) IL-4 is only used in a preventative administration, meaning the animals were pre-treated with cytokines before disease induction. It is true that we don’t have the individual cytokines in these experiments, and due to the fact that this work hasn’t been funded in nearly 2 years now, we are unable to complete those experiments. However, we believe that the data remains valuable despite the lack of individual cytokine conditions, but have reduced the “footprint” of this data to a single graph (Figure 6G) and have pointed to the caveats of our interpretation in the Results text. We believe that the data should remain in the manuscript because it demonstrates IL10-dependent prophylactic efficacy, but if the reviewers and editors feel strongly otherwise, we are willing to simply omit this single graph in a final revision. While it does add to the story, it is not necessary to make the conclusions reported.

7) In Figure 7, the EAE experiments lack comparison between IL-2/IL-4 and either of these cytokines alone. IL-2 in particular has been used in many EAE experiments with varied effects, without some indication that the IL-2/IL4 combination has differential effects to the separate cytokines. Any of this data is hardly interpretable.Previous article suggested that systemic administration of IL-4 and IL-2 alone improves EAE, possibly through DCs and Treg cells, respectively. Is there data to suggest the in vivo phenotype observed in Figure 7 is a synergy with IL-4 and IL-2? Or are they offering similar degree of protection against EAE disease development?

We added a new experiment comparing IL-2/IL-4 and the individual cytokines alone in the MOG model of EAE (Figure 7A-B). We found that combined IL-2/IL-4 has therapeutic efficacy, and IL-2 and IL-4 alone do not reduce EAE severity. The important distinction between our data and published findings using each cytokine is the dosing and treatment schedules:

1) IL-2 has been shown to limit the severity of EAE (PMID: 22954711) at a treatment regimen of 10,000 IU/dose, 3x per day i.v. for 3 days, totaling to 90,000 IU over the course of the trial. Our experiment entails administering a mere 2,500 IU of IL-2 over the course of the trial.

2) The same came be said about IL-4. We administered 16.24 ng of IL-4 per dose, every other day. The study by Racke et al. that is commonly cited administered 1ug of IL-4 every 8 hours on days 0-11 or 6-11 (doi: 10.1084/jem.180.5.1961). At the low doses we used, IL-2 and IL-4 alone were not effective at suppressing disease, however in conjunction with the other, the dual cytokine cocktail significantly suppressed disease through synergistic action.

Do conventional T cells express IL-10 in asthma model and EAE model in vivo? Would they attribute to the loss of effect in IL-10^-/-^ model?

We ran the analyses and saw that while conventional T cells do express some amounts of IL-10 in the surveyed tissues (lung and spleen for HDM-induced asthma, and CNS for EAE), the expression levels do not increase with IL-2/IL-4 treatment, in either naïve or challenged mice (Figure 6—figure supplement 1B, Figure 7—figure supplement 3A). Combined with the results of IL-2/IL-4 lacking efficacy in the new IL-10 cKO asthma and EAE trials, the IL-10 produced by Tregs appears to be the essential component (Figure 6D-E, Figure 7I-J).

8) Several other reports have looked at the role of IL-4 in Treg function and demonstrated its role in supporting Treg proliferation and suppressive function in both mice and humans. The current Discussion should include a greater overview of the current literature.Thornton, Piccirillo and Shevach, 2004; Pace, Pioli and Doria, 2005; Yang et al., 2017.

Thank you. We have edited the manuscript to include a discussion of these references.